# Descriptive analysis of dental X-ray images using various practical methods: A review

Anuj Kumar, Harvendra Singh Bhadauria and Annapurna Singh

Department of Computer Science & Engineering, Govind Ballabh Pant Institute of Engineering & Technology, Ghurdauri, Pauri Garhwal, Uttarakhand, India

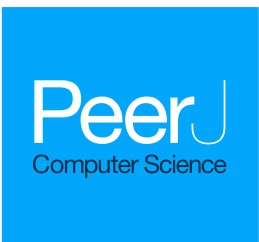

## ABSTRACT

In dentistry, practitioners interpret various dental X-ray imaging modalities to identify tooth-related problems, abnormalities, or teeth structure changes. Another aspect of dental imaging is that it can be helpful in the field of biometrics. Human dental image analysis is a challenging and time-consuming process due to the unspecified and uneven structures of various teeth, and hence the manual investigation of dental abnormalities is at par excellence. However, automation in the domain of dental image segmentation and examination is essentially the need of the hour in order to ensure error-free diagnosis and better treatment planning. In this article, we have provided a comprehensive survey of dental image segmentation and analysis by investigating more than 130 research works conducted through various dental imaging modalities, such as various modes of X-ray, CT (Computed Tomography), CBCT (Cone Beam Computed Tomography), etc. Overall state-of-the-art research works have been classified into three major categories, *i.e.*, image processing, machine learning, and deep learning approaches, and their respective advantages and limitations are identified and discussed. The survey presents extensive details of the state-of-the-art methods, including image modalities, pre-processing applied for image enhancement, performance measures, and datasets utilized.

Corresponding author
Anuj Kumar,
dranujdhiman@gmail.com

## INTRODUCTION

Dental X-ray imaging (DXRI) has been developed as the foundation for dental professionals across the world because of the assistance provided in detecting the abnormalities present in the teeth structures (*Oprea et al., 2008*). For dentists, radiography imparts a significant role in assisting imaging assessment in providing a thorough clinical diagnosis and dental structures preventive examinations (*Molteni, 1993*). However, to analyze a dental X-ray image, researchers primarily use image processing methods to extract the relevant information. Image segmentation is the most widely used image-processing technique to analyze medical images and help improve computer-aided medical diagnosis systems (*Li et al., 2006*; *Shah et al., 2006*).

Furthermore, manual examination of a large collection of X-ray images can be time-consuming because visual inspection and tooth structure analysis have an abysmal sensitive rate; therefore, human screening may not identify a high proportion of caries (*Olsen et al., 2009*). In most cases, the automatic computerized tool that can help the

investigation process would be highly beneficial (*Abdi, Kasaei & Mehdizadeh, 2015*; *Jain & Chauhan, 2017*). Dental image examination involved various stages consisting of image enhancement, segmentation, feature extractions, and identification of regions, which are subsequently valuable for detecting cavities, tooth fractures, cysts or tumors, root canal length, and teeth growth in children (*Kutsch, 2011*; *Purnama et al., 2015*). Also, various studies revealed that analysis of dental imaging modalities is beneficial in applications like human identification, age estimation, and biometrics (*Nomir & Abdel-Mottaleb, 2007*; *Caruso, Silvestri & Sconfienza, 2013*).

At present, deep learning (DL) and machine learning (ML) techniques have gained huge momentum in the field of DXRI analysis. Deep learning frameworks, well-known as convolutional neural networks (CNNs), are primarily employed for processing large and complex image datasets because they can obtain multiple features from obfuscated layers (*Schmidhuber, 2015*; *Hwang et al., 2019*). Many studies that used pre-trained networks like Alexnet, VGG, GoogLeNet, and Inception v3 found that they performed well in general. On the other hand, CNN networks tend to develop from shallow layer networks to broader or problem-specific self-made or complicated networks.

Recently, numerous machine learning approaches have been proposed by researchers to improve dental image segmentation and analysis performance. Deep learning and artificial intelligence techniques are remarkably successful in addressing the challenging segmentation dilemmas presented in various studies (*Hatvani et al., 2018*; *Lee et al., 2018a*; *Yang et al., 2018*; *Hwang et al., 2019*; *Khanagar et al., 2021*), so we can foresee a whirlwind of inventiveness and lines of findings in the coming years, based on achievements that recommend machine learning models concerning semiotic segmentation for DXRI.

In the existing surveys (*Rad et al., 2013*; *Schwendicke et al., 2019*), various techniques and methods have been discussed for DXRI. In *Rad et al. (2013)*, segmentation techniques are divided into three classes: pixel-based, edge-based, and region-based, and further classified into thresholding, clustering boundary-based, region-based, or watershed approaches. However, there is no discussion on enhancement techniques, image databases used, and modalities used for DXRI. Furthermore, after the *Rad et al. (2013)* survey, a large number of approaches have been introduced by researchers. Next, a review of dental image diagnosis using convolution neural network is presented by *Schwendicke et al. (2019)*, focusing on diagnostic accuracy studies that pitted a CNN against a reference test, primarily on routine imagery data. It has been observed that in the previous surveys, a thorough investigation of traditional image processing, machine learning, and deep learning approaches is missing.

Being an emerging and promising research domain, dental X-ray imaging requires a comprehensive and detailed survey of dental image segmentation and analysis to diagnose and treat various dental diseases. In this study, we have made the following contributions that are missing in the previous surveys: First, we have imparted various studies from 2004 to 2020 covering more than 130 articles and is almost double than

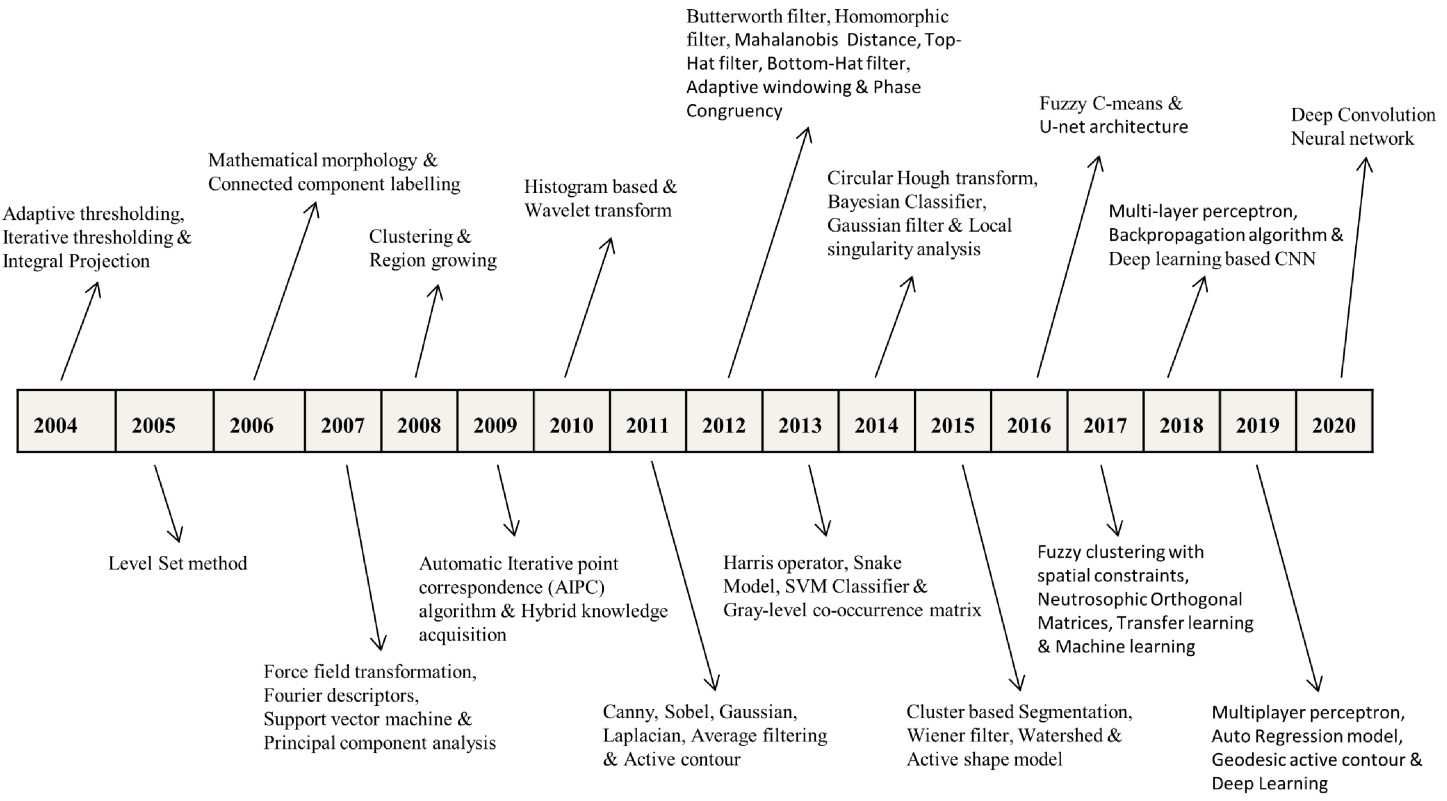

**Figure 1 Selected benchmarks at various years for dental imaging methods.**

previous surveys given by *Rad et al. (2013)* and *Schwendicke et al. (2019)*. Second, we have presented X-ray pre-processing techniques, traditional image analysis approaches, machine learning, and deep learning advancements in DXRI. Third, specific image modality (such as periapical, panoramic, bitewing and CBCT, etc.) based methods are categorized. At last, performance metrics and dataset descriptions are investigated up to a great extent. Also, specific benchmarks in the advancement of DXRI methods are represented in Fig. 1.

## A brief about dental imaging modalities

Dental imaging modalities give insights into teeth growth, bone structures, soft tissues, tooth loss, decay and also helps in root canal treatment (RCT), which is not visible during a dentist's clinical inspection. Dental imaging modalities are mainly categorized as intra-oral and extra-oral X-rays. In dentistry, these images are frequently used for medical diagnosis (*Abrahams, 2001*; *Caruso, Silvestri & Sconfienza, 2013*). Various dental imaging modalities categorization based on intra-oral and extra-oral are presented in Fig. 2.

Dental radiographs can discover problems in the mouth, jaws, teeth, bone loss, fractures, cysts at an early stage. X-rays can help in finding issues that can not be visualized with an oral assessment. Identifying and diagnosing problems at the earliest stage can save you from root canal treatment and other serious issues.

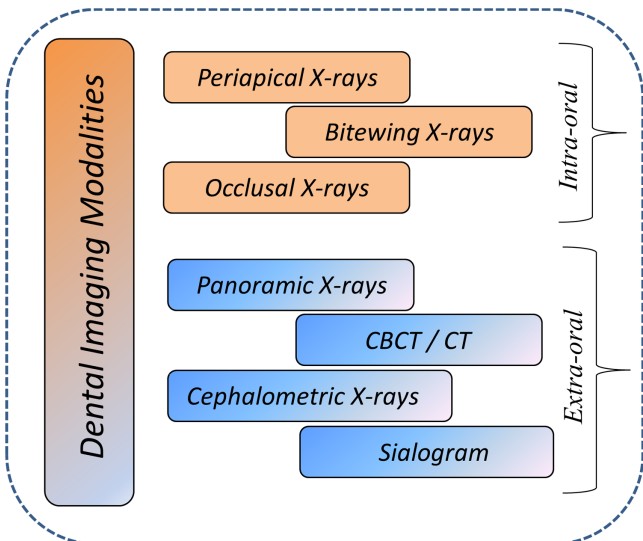

**Figure 2  Overview of dental imaging modalities.**       

### Types of dental radiography

**Intra-oral radiography.** An X-ray film is kept in the mouth to capture the X-ray picture, which comprises all the specific details about teeth arrangement, root canal infection, and identifying caries. Categories of intra-oral X-ray images are:

- **Periapical images.** It provides information of root and surrounding bone areas containing three to four teeth in the single X-ray image.
- **Bitewing images.** It generally helps in detecting the information of upper and lower tooth arrangements, and an X-ray beam shows the dentist how these teeth are arranged with one another and how to spot a cavity between teeth. Bitewing X-rays may also be used to ensure that a crown is fitted correctly (a tooth-enclosing cap) or tooth restoration is done accurately. It can also detect rotting or damaged fillings.
- **Occlusal images.** Occlusal X-rays provide insight into the mouth's base, revealing the upper or lower jaw's bite. They place a strong emphasis on children's tooth development and placement.

**Extra-oral radiography.** An X-ray picture is taken from outside the mouth to capture the entire skull and jaws region. Extra-oral X-rays are classified into many types.

- **Panoramic X-rays.** X-rays are full-sized and capture the overall tooth structure. Also, the pictures provide information about the skull and jaw. These images are mainly used to examine fractures, trauma, jaws diseases, pathological lesions and evaluate the impacted teeth.
- **Cephalometric X-rays.** Also called ceph X-ray, it depicts the jaw's whole part, including the head's entire side. It is employed in both dentistry and medicine for diagnosis and clinical preparation purposes.

- *Sialogram.* It uses a substance that is infused into the salivary glands to make them visible on X-ray film. Doctors may recommend this check to ensure problems with the salivary glands, such as infections or Sjogren's syndrome signs (a symptom condition identified by sore mouth and eyes; this condition may cause tooth decay).
- *Computed tomography (CT).* It is an imaging technique that gives insights into 3-D internal structures. This kind of visualization is used to identify maladies such as cysts, cancers, and fractures in the face's bones.
- *Cone-beam computed tomography (CBCT)* generates precise and high-quality pictures. Cone beam CT is an X-ray type that generates 3D visions of dental formations, soft tissues, nerves, and bones. It helps in guiding the tooth implants and finding cyst and tumefaction in the mouth. It can also find issues in the gum areas, roots, and jaws structures. Cone beam CT is identical to standard dental CT in several respects.

In this study, various articles considered in which the researchers proposed techniques that are extensively applied to periapical, bitewing, panoramic, CT, CBCT, and photographic color images. Digital X-ray imaging is currently gaining traction as a new research area with expanding applications in various fields.

## Challenges faced by doctors in analyzing dental X-ray images

Dental practitioners used X-ray radiographs to examine dental anatomy and to determine the care strategy for the patient. Because of a lack of resources, X-ray interpretations rely more on the doctor's expertise, and manual examination is complex in dentistry (*Wang et al., 2016*). Therefore, computer-aided systems are introduced to reduce complexity and make the identification process easy and fast. Computer-aided systems are becoming more powerful and intelligent for identifying abnormalities after processing medical images (such as X-rays, Microscopic images, Ultrasound images, and MRI images). Healthcare decision support systems were developed to provide technical guidance to clinical decision-making experts in the healthcare field (*Mendonça, 2004*). These systems help identify and treat the earliest symptom of demineralization of tooth caries, root canal, and periodontal diseases.

This paper explores the potential computational methods used for developing computer-aided systems, identifies the challenges faced in their implementation, and provides future directions (*Amer & Aqel, 2015*; *Wang et al., 2016*). Automatic detection of abnormalities, anomalies, and abrupt changes in teeth structures is a big challenge for researchers. In this study, some of the tooth-related problems are imparted, which is still a challenge for researchers to develop expert systems. We have worked with some of the dental practitioners to understand the common problems. These problems are significantly related to cavities (or caries), root canal treatment (RCT), cysts, teeth implants, and teeth growth. Working in collaboration with dentists helps computer science professionals and researchers to design & develop models that can solve dentist's problems during examination.

The dental X-ray image analysis methods can be categorized in several categories: region growing techniques, edge detection methods, thresholding based, clustering

techniques, level set, and active contour, etc., are presented in 'Image processing methods for dental image analysis' (*Mahoor & Abdel-Mottaleb, 2004*; *Zhou & Abdel-Mottaleb, 2005*; *Nomir & Abdel-Mottaleb, 2005*, *2007*; *Gao & Chae, 2008*; *Oprea et al., 2008*; *Patanachai, Covavisaruch & Sinthanayothin, 2010*; *Harandi & Pourghassem, 2011*; *Hu et al., 2014*; *Amer & Aqel, 2015*; *Zak et al., 2017*; *Avuçlu & Bacsçiftçi, 2020*) (*Rad et al., 2015*; *Tuan, Ngan & Son, 2016*; *Poonsri et al., 2016*; *Son & Tuan, 2016*, *2017*; *Ali et al., 2018*; *Alsmadi, 2018*; *Obuchowicz Rafałand Nurzynska et al., 2018*; *Tuan et al., 2018*; *Fariza et al., 2019*; *Kumar, Bhadauria & Singh, 2020*).

Conventional machine learning methods considering: back propagation neural network (BPNN), artificial neural network (ANN), support vector machine (SVM), Random forest regression-voting constrained local model (RFRV-CLM), Hybrid learning algorithms are presented in 'Conventional machine learning algorithms for dental image analysis' (*Nassar & Ammar, 2007*; *Fernandez & Chang, 2012*; *Pushparaj et al., 2013*; *Prakash, Gowsika & Sathiyapriya, 2015*; *Bo et al., 2017*; *Yilmaz, Kayikcioglu & Kayipmaz, 2017*; *Vila-Blanco, Tomás & Carreira, 2018*). Also, considering Deep learning architectures, *i.e.*, Conventional CNN and transfer learning, GoogLeNet Inception v3, AlexNet, Mask R-CNN model, ResNet-101, six-Layer DCNN, U-net architecture, and LightNet and MatConvNet, etc., are highlighted in 'Deep learning techniques for dental image analysis' (*Imangaliyev et al., 2016*; *Miki et al., 2017b*, *2017a*; *Oktay, 2017*; *Prajapati, Nagaraj & Mitra, 2017*; *Rana et al., 2017*; *Srivastava et al., 2017*; *Chu et al., 2018*; *Lee et al., 2018a*, *2019*; *Egger et al., 2018*; *Torosdagli et al., 2018*; *Yang et al., 2018*; *Zhang et al., 2018*; *Hatvani et al., 2018*; *Jader et al., 2018*; *Karimian et al., 2018*; *Kim et al., 2019*; *Murata et al., 2019*; *Tuzoff et al., 2019*; *Fukuda et al., 2019*; *Hiraiwa et al., 2019*; *Banar et al., 2020*; *Singh & Sehgal, 2020*; *Geetha, Aprameya & Hinduja, 2020*).

## Contribution

DXRI analysis is an evolving and prospective research field, but still, there is a lack of systematic study available except for one or two studies. In this study, we have made significant contributions as follows:

1. A comprehensive survey consisting of more than 130 articles related to dental imaging techniques for the last 15 years is presented.
2. Overall state-of-the-art research works have been classified into three major categories, *i.e.*, image processing, machine learning, and deep learning approaches, and their respective advantages and limitations are identified and discussed.
3. A comprehensive review of dental imaging methods provided in terms of various performance metrics.
4. At last, a review of dental X-ray imaging datasets used for implementation and generation.

The rest of the review is structured as follows. The methodology is discussed in 'Methodology'. Various performance metrics are presented in 'Performance Measures'.

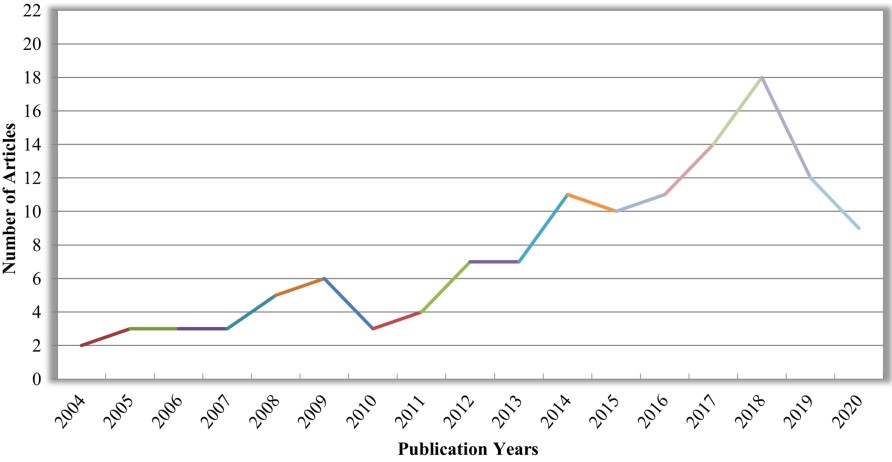

**Figure 3 Number of research articles as per publication years in DXRI.**

**Table 1 Number of articles categorized based on imaging modalities.**

| Image modalities | Number of articles published |
|---|---|
| Periapical X-ray images | 30 |
| Bitewing X-ray images | 11 |
| Panoramic X-ray images | 39 |
| CBCT or CT images | 13 |
| Photographic color images | 06 |
| Hybrid dataset | 19 |
| Image dataset not defined | 07 |

DXRI datasets are given in 'Dataset Description'. At last, the conclusion is given in 'Conclusion'.

## METHODOLOGY

In this survey, 130 research articles from 2004 to 2020 have been reviewed, as shown in Fig. 3, covering almost all research articles from different online digital libraries like Springer, Elsevier, IEEE, and Google Scholar. These articles are conferences, Book chapters, peer-reviewed and reputed journals in computer science and digital dental imaging. A total number of articles deliberating various imaging modalities: Periapical, Bitewing, Panoramic, Hybrid, CT or CBCT, Photographic color teeth images, and undefined datasets are given in Table 1. Methods are categorized as image processing techniques in 'Image processing methods for dental image analysis', conventional machine learning methods are given in 'Conventional machine learning algorithms for dental image analysis', and deep learning approaches are provided in 'Deep learning techniques for dental image analysis'. Also, methods are characterized based on imaging modalities (Periapical X-rays, Bitewing X-rays, Panoramic X-rays, CBCT or CT images, etc.), and DXRI methods taxonomy is given in Fig. 4.

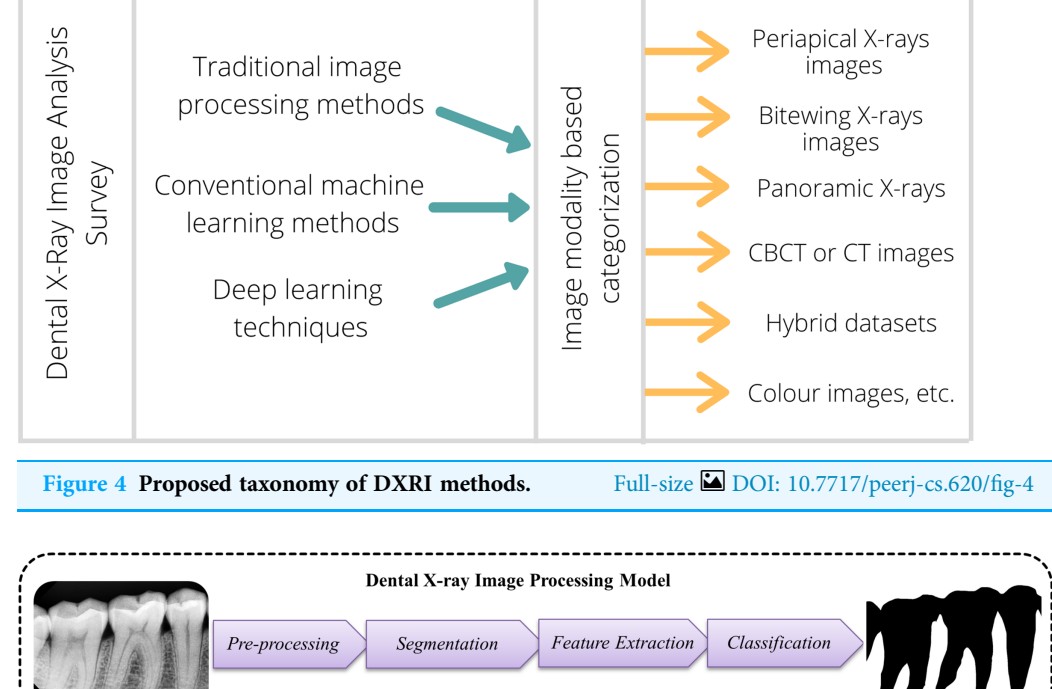

**Figure 4 Proposed taxonomy of DXRI methods.**

**Figure 5 Traditional model used for dental image segmentation and classification.**

The research incorporated in this comprehensive review primarily focused on medical image processing and artificial intelligence for the detection and examination of the tooth cavity, periodontal disease recognition, tooth arrangement and numbering, root canal detection, periapical lesions detection, salivary gland disease diagnosis, cyst detection, osteoporosis detection, the progress of deciduous teeth, analysis of cephalometric landmarks and fracture identification, etc.

## Image processing methods for dental image analysis

The research adopts various image processing strategies for dental imaging to investigate the structures of teeth, caries, and abnormalities to help dental practitioners for the appropriate diagnosis. It involves various pre-processing, segmentation, and classification approaches to make an automatic dental identification system that makes doctor's work more accessible, unambiguous, and faster. A simple traditional model used for dental image processing is given in Fig. 5.

### Pre-processing techniques

Dental imaging consists of different image modalities, where X-rays are the most common medical imaging method used to classify bone and hard tissues. In dentistry, imaging modalities help identify fractures, teeth structures, jaws alignment, cyst, and bone loss, which has become tremendously popular in dental imaging (*Goyal, Agrawal & Sohi, 2018*). Noise level, artifacts, and image contrast are vital values that control an image's overall

quality. The image quality obtained depends on varying factors such as the dynamic range of the sensors, the lighting conditions, distortion, and the artifact examined (*Sarage & Jambhorkar, 2012*). Interpretation of a low-resolution image is often a complex and time-consuming process. Pre-processing techniques enhance the quality of low-resolution images, which corrects the spatial resolution and local adjustment to improve the input image's overall quality (*Hossain, Alsharif & Yamashita, 2010*). Moreover, enhancement and filtering methods improve the overall image quality parameters before further processing. In Table 2, pre-processing techniques are addressed to recuperate the quality of dental images.

Contrast stretching, Grayscale stretching, Log transformation, Gamma correction, Image negative, and Histogram equalization methods are standard enhancement methods to improve the quality of medical images. X-rays are typically grayscale pictures with high noise rates and low resolution. Thus, the image contrast and boundary representation are relatively weak and small (*Ramani, Vanitha & Valarmathy, 2013*). Extracting features from these X-rays is quite a difficult task with very minimal details and a low-quality image. By adding specific contrast enhancement techniques significantly improves image quality. So that segmentation and extraction of features from such images can be performed more accurately and conveniently (*Kushol et al., 2019*). Therefore, a contrast stretching approach has been widely used to enhance digital X-rays quality (*Lai & Lin, 2008*; *Vijayakumari et al., 2012*; *Berdouses et al., 2015*; *Purnama et al., 2015*; *Avuçlu & Bacsçiftçi, 2020*). Adaptive local contrast stretching makes use of local homogeneity to solve the problem of over and under enhancement. One of the prominent methods to refine the contrast of the image is histogram equalization (HE) (*Harandi & Pourghassem, 2011*; *Menon & Rajeshwari, 2016*; *Obuchowicz Rafałand Nurzynska et al., 2018*; *Banday & Mir, 2019*). HE is the way of extending the dynamic range of an image histogram and it also causes unrealistic impacts in images; however, it is very effective for scientific pictures i.e., satellite images, computed tomography, or X-rays. A downside of the approach is its indiscriminate existence. This can increase ambient noise contrast while reducing the useful quality features of an image.

On the other hand, filtering methods applied to medical images help to eradicate the noise up to some extent. Gaussian, Poisson, and Quantum noise are different types of noise artifacts usually found in X-Rays & CTs, particularly when the image is captured (*Razifar et al., 2005*; *Goyal, Agrawal & Sohi, 2018*). The noise-free images achieve the efficiency to get the best result and improve the test's precision. If we try to minimize one class of noise, it may disrupt the other. Various filters have been used to achieve the best potential outcome for the irregularities present in dental images like Average filter, Bilateral filter, Laplacian filter, Homomorphic filter, and Butterworth filter, Median Gaussian filter, and Weiner filter. In recent studies, various filtering techniques used by researchers but widely used filtering methods are Gaussian filter and the median filter, which shows the best result (*Benyó et al., 2009*; *Prajapati, Desai & Modi, 2012*; *Nuansanong, Kiattisin & Leelasantitham, 2014*; *Razali et al., 2014*; *Datta & Chaki, 2015a,b*; *Rad et al., 2015*; *Tuan, Ngan & Son, 2016*; *Jain & Chauhan, 2017*; *Alsmadi, 2018*). However, the drawback of the median filter is that it degrades the boundary details. Whereas the

**Table 2 Pre-processing methods used for dental imaging modality.**

| Author & Year | Enhancement/Noise removal technique |
| --- | --- |
| **Methods used for Bitewing X-ray** | |
| (*Lai & Lin, 2008*) | Adaptive local contrast stretching is used to make the tooth region smoother after that, adaptive morphological enhancement is applied to improve the texture values. |
| (*Prajapati, Desai & Modi, 2012*) | A median filter is used to eradicate picture impulse noise. |
| (*Mahoor & Abdel-Mottaleb, 2004*; *Zhou & Abdel-Mottaleb, 2005*; *Huang et al., 2012*) | Top hat and bottom hat filters are applied where the teeth become brightened, and the bone and shadow regions obscured. |
| (*Pushparaj, Gurunathan & Arumugam, 2013*) | Butterworth high pass filter used with a homomorphic filter. In which homomorphic filter compensates the effect of non-uniform illumination. |
| **Methods used for Periapical X-ray** | |
| (*Harandi & Pourghassem, 2011*) | Histogram equalization and noise reduction using wavelets, and also make use of spatial filters like Laplacian filter. |
| (*Lin, Huang & Huang, 2012*) | Average filter with 25 * 25 mask then histogram equalization is used. |
| (*Nuansanong, Kiattisin & Leelasantitham, 2014*) | Gaussian spatial filter with kernel size 5 * 5 and sigma value 1.4 is fixed. |
| (*Lin et al., 2014*) | Enhancement is done by combining adaptive power law transformation, local singularity, and bilateral filter. |
| (*Rad et al., 2015*) | Median filtering is applied to enhance the images |
| (*Purnama et al., 2015*) | Contrast stretching used to improve the X-ray quality so that it can be easily interpreted and examined correctly |
| (*Jain & Chauhan, 2017*) | Gaussian filtering employed to make a more smoothed gradient nearby the edges also helps in reducing noise. |
| (*Obuchowicz Rafałand Nurzynska et al., 2018*) | Histogram equalization (HEQ) and a statistical dominance algorithm (SDA) are initiated. |
| (*Singh & Agarwal, 2018*) | Median filtering is used to lower noise, and an unsharp marking filter is used to enhance the high-frequency component. |
| (*Datta, Chaki & Modak, 2019*) | Local averaging is used to eliminate noisy features. |
| (*Kumar, Bhadauria & Singh, 2020*) | The guided filter is applied with a window size of 3 * 3 and is cast-off towards calculating output pixel size. |
| **Methods used for Panoramic X-rays** | |
| (*Frejlichowski & Wanat, 2011*) | Some basic filters are added to select pyramid layers, including sharpening filter and contrast adjustment before image recomposition. |
| (*Vijayakumari et al., 2012*) | Block analysis and contrast stretching applied. |
| (*Pushparaj et al., 2013*) | A combination of the Butterworth bandpass filter and the homomorphic filter is used to enhance the edges and illumination. |
| (*Razali et al., 2014*) | Canny edge detection is applied, where the gaussian filter is used to eliminate the noise. |
| (*Banu et al., 2014*) | Image inverse and contrast stretching procedures have been used to identify the region of interest. |
| (*Amer & Aqel, 2015*) | Contrast enhancement with intensity transformations is used to improve the segmentation procedure. |
| (*Poonsri et al., 2016*) | Image enhancement using adaptive thresholding (*Bradley & Roth, 2007*). |
| (*Veena Divya, Jatti & Revan Joshi, 2016*) | The image contrast is balanced to enhance the picture's appearance and to visualize the cyst or tumor. |
| (*Zak et al., 2017*) | A combination of top hat/bottom hat filter and adaptive power-law transformation(APLT) is used to enhance images. |
| (*Alsmadi, 2018*) | Speckle noise is reduced by using a median filter. |
| (*Divya et al., 2019*) | Negative transformation applied and caries identified by using the difference of contrast improved Image and image negative. |

| Table 2 (continued) | |
| --- | --- |
| **Author & Year** | **Enhancement/Noise removal technique** |
| (*Banday & Mir, 2019*) | Adaptive histogram equalization and median filtering are combinedly applied. |
| (*Fariza et al., 2019*) | Dental X-ray image is processed using CLAHE, and gamma correction is done to improve the contrast. |
| (*Avuçlu & Bacsçiftçi, 2020*) | Median softening filter applied after contrast stretching. |
| **Methods used for hybrid dataset** | |
| (*Said et al., 2006*) | Internal noise is reduced by closing top-hat transformation, which is described by subtracting the picture from its morphological closure. |
| (*Tuan, Ngan & Son, 2016*) | Background noise is minimized using a Gaussian filter; then, a Gaussian(DoG) filter is used to measure the gradient along the x and y-axis. |
| **Methods used for color images** | |
| (*Ghaedi et al., 2014*) | A contrast enhancement focused on the histogram is introduced to the gray-level Image. |
| (*Datta & Chaki, 2015a*) | Denoising is done by using a wiener filter. |
| (*Datta & Chaki, 2015b*) | A Wiener filter is applied to eliminate the blurring effect and additive noise. |
| (*Berdouses et al., 2015*) | Gray level transformation performed. |
| **Methods used for CBCT & CT** | |
| (*Benyó et al., 2009*) | Image with high-frequency noise are enhanced by applying a median filter |
| (*Ji, Ong & Foong, 2014*) | Initially, the intensity range was adjusted, followed by Gaussian filtering with a standard deviation to suppress noise. |

Gaussian filter performs best in peak detection, the limitation is that it reduces the picture's information.

### Dental image segmentation approaches used for different imaging modalities

DXRI segmentation is an essential step to extract valuable information from various imaging modalities. In dentistry, segmentation faces more difficulties than other medical imaging modalities, making the segmentation process more complicated or challenging. Here, the problems faced by researchers in analyzing dental X-ray images and the purpose of segmentation are given in Fig. 6. The segmentation process refers to the localization of artifacts or the boundary tracing, analysis of structure, etc. Human eyes quickly distinguish objects of interest and remove them from the background tissues, but it is a great challenge in developing algorithms.

Furthermore, image segmentation has applications distinct from computer vision; it is often used to extract or exclude different portions of an image. General dental image segmentation methods are categorized as thresholding-based, contour or snake models, level set methods, clustering, and region growing (*Rad et al., 2013*). Moreover, there has been a significant number of surveys presented by various authors (*Rad et al., 2013*; *Sharma, Rana & Kundra, 2015*). However, none of them categorized the methods based on dental imaging modalities. Various segmentation and classification techniques are discussed and reviewed in this article, considering multiple dental imaging modalities. In the field of dental imaging, the choice of selecting a correct algorithm for the particular image dataset is most important. This study explores image processing techniques explicitly applied for dental imaging modalities, as given in Table 3.

## Purpose of Dental Image Segmentation

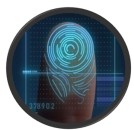 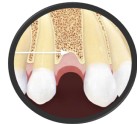 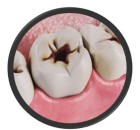 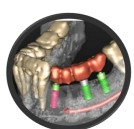 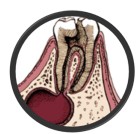 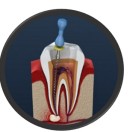

| Biometrics (Human identification) | Bone loss & Teeth gap areas | Caries detection | Computer guided treatment | Cyst or Tumour extraction | Root canal treatment diagnosis |

## Problems in Dental Image Segmentation

| Poor image quality due to presence of noise. | Irregular shape of object | Intensity variations in X-Ray | Proper selection of methods for the application | Limitation of capturing devices | Lack of availability of datasets |

**Figure 6 Purpose of segmentation & problems in dental imaging.**

*Bitewing X-rays* are widely used by researchers for the application of human identification and biometrics. Human identification is achieved by applying adaptive thresholding, iterative thresholding, and region-growing approaches. Afterwards, image features are extracted to archive and retrieve dental images used for human identification (*Mahoor & Abdel-Mottaleb, 2004*, *2005*; *Nomir & Abdel-Mottaleb, 2005*, *2007*, *2008*; *Zhou & Abdel-Mottaleb, 2005*). In *Huang et al. (2012)*, missing tooth locations were detected with an adaptive windowing scheme combined with the isolation curve method, which shows the accuracy rate higher than (*Nomir & Abdel-Mottaleb, 2005*). In *Pushparaj, Gurunathan & Arumugam (2013)*, primarily aimed at estimating the shape of the entire tooth. In which segmentation is performed by applying horizontal and vertical integral projection. In addition, teeth boundary was estimated using the fast connected component labeling algorithm, and lastly, Mahalanobis distance is measured for the matching.

*Periapical X-rays* help in clinical diagnosis considering dental caries and root canal regions by applying various image processing techniques (*Oprea et al., 2008*). Many times dentists use periapical X-ray images to spot caries lesions from dental X-rays. Regardless of human brain vision, it is often hard to correctly identify caries by manually examining the X-ray image. Caries detection methods for periapical X-rays have been used iteratively to isolate the initially suspected areas. Then, separated regions are subsequently analyzed. In *Rad et al. (2015)*, automatic caries was identified by applying segmentation using k-means clustering and feature detection using GLCM. However, it shows image quality issues in some cases, and because of these issues, tooth detection may give a false result. On the other hand, (*Singh & Agarwal, 2018*) applied color masking techniques to mark the curios lesions to find the percentage value of the affected area.

Another approach is given by (*Osterloh & Viriri, 2019*) mainly focused on upper and lower jaws separation with the help of thresholding and integral projection, and the

**Table 3 Review findings of the image processing techniques using different imaging modalities.**

| Author & Year | Relevant review findings | Total images | Detection/ Identification |
|---|---|---|---|
| **Imaging modality: Bitewing X-rays** | | | |
| (*Mahoor & Abdel-Mottaleb, 2004*) | For segmentation, adaptive thresholding methods is being used, then features are extracted, and teeth numbering is done using the Bayesian classification technique. | 50 | Teeth numbering |
| (*Zhou & Abdel-Mottaleb, 2005*) | Proposed segmentation using a window-based adaptive thresholding scheme and minimum Hausdorff distance used for matching purposes. | Training =102 images Testing = 40 images | Human identification |
| (*Nomir & Abdel-Mottaleb, 2005*) | Results are improved by using a signature vector in conjunction with adaptive and iterative thresholding. | 117 | Human identification |
| (*Nomir & Abdel-Mottaleb, 2007*) | Iterative followed by adaptive thresholding used for the segmentation and features extracted using fourier descriptors after forcefield transformation then matching is done by using euclidian distance | 162 | Human identification |
| (*Lai & Lin, 2008*) | The B-spline curve is used to extract intensity and texture characteristics for K-means clustering to locate the bones and teeth contour. | N.A | Teeth detection |
| (*Nomir & Abdel-Mottaleb, 2008*) | The procedure starts with an iterative process guided by adaptive thresholding. Finally, the Bayesian framework is employed for tooth matching. | 187 | Human identification |
| (*Harandi, Pourghassem & Mahmoodian, 2011*) | An active geodesic contour is employed for upper and lower jaws segmentation. | 14 | Jaw identification |
| (*Huang et al., 2012*) | An adaptive windowing scheme with isolation-curve verification is used to detect missing tooth regions. | 60 | Missing teeth detection |
| (*Prajapati, Desai & Modi, 2012*) | A region growing technique is applied to the X-rays to extract the tooth; then, the content-based image retrieval (CBIR) technique is used for matching purposes. | 30 | Human identification |
| (*Pushparaj, Gurunathan & Arumugam, 2013*) | The tooth area's shape is extracted using contour-based connected component labeling, and the Mahalanobis distance (MD) is measured for matching. | 50 | Person identification |
| **Imaging modality: Periapical X-rays** | | | |
| (*Huang & Hsu, 2008*) | Binary image transformations, thresholding, quartering, characterization, and labeling were all used as part of the process. | 420 | Teeth detection |
| (*Oprea et al., 2008*) | Simple thresholding technique applied for segmentation of caries. | N.A | Caries detection |
| (*Harandi & Pourghassem, 2011*) | Otsu thresholding method with canny edge detection is used to segment the root canal area. | 43 | Root canal detection |
| (*Lin, Huang & Huang, 2012*) | The lesion is detected using a variational level set method after applying otsu's method. | 6 | Lesion detection |
| (*Sattar & Karray, 2012*) | Phase congruency based approach is used to provide a framework for local image structure + edge detection | N.A | Teeth detection |
| (*Niroshika, Meegama & Fernando, 2013*) | Deformation and re-parameterize are added to the contour to detect the tooth comer points. | N.A | Teeth detection |
| (*Ayuningtiyas et al., 2013*) | Dentin and pulp are separated using active contour, and qualitative analysis is conducted using the dentist's visual inspection, while quantitative testing is done by measuring different statistic parameters. | N.A | Tooth detection |
| (*Nuansanong, Kiattisin & Leelasantitham, 2014*) | Canny edge detection was initially used, followed by an active contour model with data mining (J48 tree) and integration with the competence path. | Approx. 50 | Tooth detection |

(Continued)

| Author & Year | Relevant review findings | Total images | Detection/ Identification |
|---|---|---|---|
| (*Lin et al., 2014*) | The otsu's threshold and connected component analysis are used to precisely segment the teeth from alveolar bones and remove false teeth areas. | 28 | Teeth detection |
| (*Purnama et al., 2015*) | For root canal segmentation, an active shape model and thinning (using a hit-and-miss transform) were used. | 7 | Root canal detection |
| (*Rad et al., 2015*) | The segmentation is initially done using K-means clustering. Then, using a gray-level co-occurrence matrix, characteristics were extracted from the X-rays. | 32 | Caries detection |
| (*Jain & Chauhan, 2017*) | First, all parameter values defined in the snake model then initial contour points initializes, and at last canny edge detection extract the affected part. | N.A | Cyst detection |
| (*Singh & Agarwal, 2018*) | The color to mark the carious lesion is provided by the contrast limited adaptive histogram (CLAHE) technique combined with masking. | 23 | Caries detection |
| (*Rad et al., 2018*) | The level set segmentation process (LS) is used in two stages. The first stage is the initial contour creation to create the most appropriate IC, and the second stage is the artificial neural network-based smart level approach. | 120 | Caries detection |
| (*Obuchowicz Rafałand Nurzynska et al., 2018*) | K-means clustering applied considering intensity values and first-order features (FOF) to detect the caries spots. | 10 | Caries detection |
| (*Devi, Banumathi & Ulaganathan, 2019*) | The hybrid algorithm is applied using isophote curvature and the fast marching method (FMM) to extract the cyst. | 3 | Cyst detection |
| (*Datta, Chaki & Modak, 2019*) | The geodesic active contour method is applied to identify the dental caries lesion. | 120 | Caries detection |
| (*Osterloh & Viriri, 2019*) | It uses unsupervised model to extract the caries region. Jaws partition is done using thresholding and an integral projection algorithm. The top and bottom hats, as well as active contours, were used to detect caries borders. | N.A | Caries detection |
| (*Kumar, Bhadauria & Singh, 2020*) | The various dental structures were separated using the fuzzy C-means algorithm and the hyperbolic tangent gaussian kernel function. | 152 | Dental structures |
| (*Datta, Chaki & Modak, 2020*) | This method converts the X-ray image data into its neutrosophic analog domain. A custom feature called 'weight' is used for neutrosophication. Contrary to popular belief, this feature is determined by merging other features. | 120 | Caries detection |
| **Imaging Modality: Panoramic X-rays** | | | |
| (*Patanachai, Covavisaruch & Sinthanayothin, 2010*) | The wavelet transform, thresholding segmentation, and adaptive thresholding segmentation are all compared. Where, the results of wavelet transform show better accuracy as compare to others. | N.A | Teeth detection |
| (*Frejlichowski & Wanat, 2011*) | An automatic human identification system applies a horizontal integral projection to segment the individual tooth in this approach. | 218 | Human identification |
| (*Vijayakumari et al., 2012*) | A gray level co-occurrence matrix is used to detect the cyst. | 3 | Cyst detection |
| (*Pushparaj et al., 2013*) | Horizontal integral projection with a B-spline curve is employed to separate maxilla and mandible | N.A | Teeth numbering |
| (*Lira et al., 2014*) | Supervised learning used for segmentation and feature extraction is carried out through computing moments and statistical characteristics. At last, the bayesian classifier is used to identify different classes. | 1 | Teeth detection |

| Author & Year | Relevant review findings | Total images | Detection/ Identification |
|---|---|---|---|
| (Banu et al., 2014) | The gray level co-occurrence matrix is used to compute texture characteristics (GLCM) and classification results obtained in the feature space, focusing on the centroid and K-mean classifier. | 23 | Cyst detection |
| (Razali et al., 2014) | This study aims to compare the edge segmentation methods: Canny and Sobel on X-ray images. | N.A | Teeth detection |
| (Amer & Aqel, 2015) | The segmentation process uses the global Otsu's thresholding technique with linked component labeling. The ROI extraction and post-processing are completed at the end. | 1 | Wisdom teeth detection |
| (Abdi, Kasaei & Mehdizadeh, 2015) | Four stages used for segmentation: Gap valley extraction, canny edge with morphological operators, contour tracing, and template matching. | 95 | Mandible detection |
| (Veena Divya, Jatti & Revan Joshi, 2016) | Active contour or snake model used to detect the cyst boundary. | 10 | Cyst detection |
| (Poonsri et al., 2016) | Teeth identification, template matching using correlation, and area segmentation using K-means clustering are used. | 25 | Teeth detection |
| (Zak et al., 2017) | Individual arc teeth segmentation (IATS) with adaptive thresholding is applied to find the palatal bone. | 94 | Teeth detection |
| (Alsmadi, 2018) | In panoramic X-ray images that can help in diagnosing jaw lesions, the fuzzy C-means concept and the neutrosophic technique are combinedly used to segment jaw pictures and locate the jaw lesion region. | 60 | Lesion detection |
| (Dibeh, Hilal & Charara, 2018) | The methods use a shape-free layout fitted into a 9-degree polynomial curve to segment the area between the maxillary and mandibular jaws. | 62 | Jaw separation + teeth detection |
| (Mahdi & Kobashi, 2018) | Quantum Particle Swarm Optimization (QPSO) is employed for multilevel thresholding. | 12 | Teeth detection |
| (Ali et al., 2018) | A new clustering method based on the neutrosophic orthogonal matrix is presented to help in the extraction of teeth and jaws areas from panoramic X-rays. | 66 | Teeth detection |
| (Divya et al., 2019) | Textural details extracted using GLCM to classify the cyst and caries. | 10 | Dental caries & cyst extraction |
| (Banday & Mir, 2019) | Edge detection method for the segmentation then, the Autoregression (AR) model is adopted, and AR coefficients are derived from the feature vector. At last, matching is performed using euclidean distance. | 210 | Human identification |
| (Fariza et al., 2019) | For tooth segmentation, the Gaussian kernel-based conditional spatial fuzzy c-means (GK-csFCM) clustering algorithm is used. | 10 | Teeth detection |
| (Aliaga et al., 2020) | The region of interest is extracted from the entire X-ray image, and segmentation is performed using k-means clustering. | 370 | Osteoporosis detection, mandible detection |
| (Avuçlu & Bacsçiftçi, 2020) | The Image is converted to binary using Otsu's thresholding, and then a canny edge detector is used to find the object of interest. | 1,315 | Determination of age and gender |
| **Imaging modality: Hybrid dataset images** | | | |
| (Said et al., 2006) | Thresholding with mathematical morphology is performed for the segmentation. | A total of 500 Bitewing & 130 Periapical images. | Teeth detection |

(Continued)

| Author & Year | Relevant review findings | Total images | Detection/ Identification |
|---|---|---|---|
| (Li et al., 2006) | The fast and accurate segmentation approach used strongly focused on mathematical morphology and shape analysis. | A total of 500 (Bitewing and Periapical images) | Person identification |
| (Al-sherif, Guo & Ammar, 2012) | A two-phase threshold processing is used, starting with an iterative threshold followed by an adaptive threshold to binarize teeth images after separating the individual tooth using the seam carving method. | A total of 500 Bitewing & 130 Periapical images | Teeth detection |
| (Ali, Ejbali & Zaied, 2015) | The Chan-vese model and an active contour without edges are used to divide an image into two regions with piece-constant intensities. | N.A | Teeth detection |
| (Son & Tuan, 2016) | The otsu threshold procedure, fuzzy C-means, and semi-supervised fuzzy clustering are all part of a collaborative framework (eSFCM). | A total of eight & 56 Image dataset (Bitewing + Panoramic) | Teeth structures |
| (Tuan, Ngan & Son, 2016) | It uses a semi-supervised fuzzy clustering algorithm – SSFC-FS based on the Interactive Fuzzy Satisficing method. | A total of 56 (Periapical & Panoramic) | Teeth structures |
| (Son & Tuan, 2017) | Semi-supervised fuzzy clustering algorithm combined with spatial constraints (SSFC-SC) for dental image segmentation. | A total of 56 (Periapical & panoramic images) | Teeth structures |
| (Tuan et al., 2018) | Graph-based clustering algorithm called enhanced affinity propagation clustering (APC) used for classification process and fuzzy aggregation operators used for disease detection. | A total of 87 (Periapical & Panoramic) | Disease detection |
| **Imaging modality: Photographic color images** | | | |
| (Ghaedi et al., 2014) | Segmentation functions in two ways. In the first step, the tooth surface is partitioned using a region-widening approach and the Circular Hough Transform (CHT). The second stage uses morphology operators to quantify texture to define the abnormal areas of the tooth's boundaries. Finally, a random forest classifies the various classes. | 88 | Caries detection |
| (Datta & Chaki, 2015a) | The method uses a biometrics dental technique using RGB images. Segment individual teeth with water Shed and Snake's help, then afterward incisors teeth features are obtained to identify the human. | A total of 270 images dataset | Person identification |
| (Datta & Chaki, 2015b) | The proposed method introduces a method for filtering optical teeth images and extracting caries lesions followed by cluster-based Segmentation. | 45 | Caries detection |
| (Berdouses et al., 2015) | The proposed scheme included two processes: (a) identification, in which regions of interest (pre-cavitated and cavitated occlusal lesions) were partitioned, and (b) classification, in which the identified zones were categorized into one of the seven ICDAS classes. | 103 | Caries detection |
| **Imaging modality: CT & CBCT** | | | |
| (Gao & Chae, 2008) | The multi-step procedure using thresholding, dilation, connected component labeling, upper-lower jaw separation, and last arch curve fitting was used to find the tooth region. | N.A | Teeth detection |
| (Hosntalab et al., 2010) | Otsu thresholding, morphological operations, and panoramic re-sampling, and variational level set were used. Following that, feature extraction with a wavelet-Fourier descriptor (WFD) and a centroid distance signature is accomplished. Finally, multilayer perceptron (MLP), Kohonen self-organizing network, and hybrid structure are used for Classification. | 30 Multislice CT image (MSCT) dataset consists of 804 teeth | Teeth detection and Classification |
| (Gao & Chae, 2010) | An adaptive active contour tracking algorithm is used. In which the root is tracked using a single level set technique. In addition, the variational level was increased in several ways. | A total of 18 CT images | Teeth detection |

| Table 3 (continued) | | | |
|---|---|---|---|
| Author & Year | Relevant review findings | Total images | Detection/ Identification |
| (*Mortaheb, Rezaeian & Soltanian-Zadeh, 2013*) | Mean shift algorithm is used for CBCT segmentation with new feature space and is compared to thresholding, watershed, level set, and active contour techniques. | A total of two CBCT images | Teeth detection |
| (*Gao & Li, 2013*) | The volume data are initially divided into homogeneous blocks and then iteratively merged to produce the initial labeled and unlabeled instances for semi-supervised study. | N.A | Teeth detection |
| (*Ji, Ong & Foong, 2014*) | The study adds a new level set procedure for extracting the contour of the anterior teeth. Additionally, the proposed method integrates the objective functions of existing level set methods with a twofold intensity model. | A total of ten CBCT images | Teeth structure |
| (*Hu et al., 2014*) | Otsu and mean thresholding technique combinedly used to improve the segmentation. | Image dataset consists of 300 layers | Teeth detection |

learning model is employed to extract caries. This model shows better accuracy than (*Dykstra, 2008*; *Tracy et al., 2011*; *Valizadeh et al., 2015*). In *Obuchowicz Rafałand Nurzynska et al. (2018)*, k-means clustering (CLU) and first-order features (FOF) were used to show the best performance for the identification of caries. However, this approach was applied to the dataset of 10 patients with confirmed caries. A geodesic contour technique (*Datta, Chaki & Modak, 2019*) shows better computational time results than multilevel thresholding, watershed, and level set. The limitation of this approach is that it does not work well for poor-quality pictures, which leads to inappropriate feature extraction. In *Datta, Chaki & Modak (2020)*, a method reduced the computational efforts and caries region identified in optimum time. The X-ray image is processed in the neutrosophic domain to identify the suspicious part, and an active contour method is employed to detect the outer line of the carious part. The benefit of this method is that it prevents recursive iterations using neutrosophication during suspicious area detection.

The semi-automatic method for root canal length detection is proposed by *Harandi & Pourghassem (2011)* and *Purnama et al. (2015)* to help dental practitioners properly treat root canal treatment (RCT). In some studies, periapical X-rays are also used for the automatic segmentation of cysts or abscesses. *Devi, Banumathi & Ulaganathan (2019)* proposed a fully automated hybrid method that combined feature-base isophote curvature and model-based fast marching (FMM). It shows good accuracy and optimum results as compared to *Jain & Chauhan (2017)*. Furthermore, various approaches were used to automatically detect teeth structures (*Huang & Hsu, 2008*; *Sattar & Karray, 2012*; *Niroshika, Meegama & Fernando, 2013*; *Nuansanong, Kiattisin & Leelasantitham, 2014*; *Kumar, Bhadauria & Singh, 2020*).

***Panoramic X-rays*** help identify jaw fractures, the structure of jaws, and deciduous teeth. These X-rays are less detailed as compared to periapical and bitewing. It has been observed that the segmentation of panoramic X-rays using wavelet transformation shows better results than adaptive and iterative thresholding (*Patanachai, Covavisaruch & Sinthanayothin, 2010*). Another fully automatic segmentation of the teeth using the

template matching technique introduced by *Poonsri et al. (2016)* shows 50% matching accuracy results. In *Razali et al. (2014)* analyzed X-rays for the age estimations by comparing edge detection approaches. *Amer & Aqel (2015)* have suggested a method used to extract wisdom teeth using the Otsu's threshold combined with morphological dilation. Then, jaws and teeth regions are extracted using connected component labeling.

In *Mahdi & Kobashi (2018)*, it sets a multi-threshold by applying quantum particle swarm optimization to improve the accuracy. *Fariza et al. (2019)* employed a method to extract dentin, enamel, pulp, and other surrounding dental structures using conditional spatial fuzzy C-means clustering. Subsequently, the performance improved as compared to inherently used FCM approaches. *Dibeh, Hilal & Charara (2018)* separates maxillary and mandibular jaws using N-degree polynomial regression. In *Abdi, Kasaei & Mehdizadeh (2015)*, a four-step method is proposed: gap valley extraction, modified canny edge detector, guided iterative contour tracing, and template matching. However, estimating the overall performance of automated segmentation with individual results, all of which were estimated to be above 98%, clearly demonstrates that the computerized process can still be improved to meet the gold standard more precisely.

In *Veena Divya, Jatti & Revan Joshi (2016)*, active contour-based segmentation is proposed for cystic lesion segmentation and extraction to analyze cyst development behavior. The segmentation method has positive results for nonlinear background, poor contrast, and noisy image. *Divya et al. (2019)* has compared the level set method and watershed segmentation to detect cysts and lesions. The study reveals that the level set segmentation produces more predicted results for cyst/Lesion. An approach used to identify age & gender by analyzing dental images is very useful in biometrics (*Avuçlu & Bacsçiftçi, 2020*). Several other image processing techniques are used on dental images to achieve the best biometric results.

*Hybrid-dataset* is the image dataset combining different dental imaging modalities used for the analysis. *Said et al. (2006)* have used periapical & bitewing X-rays for the teeth segmentation. In this approach, the background area is discarded using an appropriate threshold, then mathematical morphology and connected component labeling are applied for the teeth extraction. This approach finds difficulty in extracting images having low contrast between teeth and bones, blurred images, etc. Another approach introduced by *Tuan, Ngan & Son (2016)*, *Son & Tuan (2017)*, *Tuan et al. (2018)* the semi-supervised fuzzy clustering method with some modification to find the various teeth and bone structures. *Ali, Ejbali & Zaied (2015)* compared CPU & GPU results after applying the Chan-Vese model with active contour without edge. It shows that GPU model implementation is several times faster than the CPU version.

*Photographic color images* are the RGB images of occlusal surfaces that are mainly useful for detecting caries and human identification (*Datta & Chaki, 2015a, 2015b*). Teeth segmentation is performed by integrating watershed and snake-based techniques on dental RGB images. Subsequently, incisors tooth features extracted for the recognition of a person. This method can segment individual teeth, lesions from caries and track the development of lesion size. This research's primary objective is to identify the caries lesions of the tooth surfaces, which benefits to improve the diagnosis. In *Ghaedi et al. (2014)*,

caries segmentation was employed using the region-widening method and circular hough transform (CHT), then morphological operations applied to locate the unstable regions around the tooth boundaries. Another fully automatic approach for the caries classification is given by *Berdouses et al. (2015)*, where segmentation separates caries lesion then after area features are extracted to assign the region to a particular class. It can be a valuable method to support the dentist in making more reliable and accurate detection and analysis of occlusal caries.

*CT & CBCT Images* provide 3D visualization of teeth and assist dental practitioners in orthodontic surgery, dental implants, and cosmetic surgeries. *Hosntalab et al. (2010)* recommended a multi-step procedure for labeling and classification in CT images. However, teeth segmentation is performed by employing global thresholding, morphological operations, region growing, and variational level sets. Another approach, a multi-step procedure, was introduced by *Mortaheb, Rezaeian & Soltanian-Zadeh (2013)* based on the mean shift algorithm for CT image segmentation of the tooth area, which results best as compare with watershed, thresholding, active contour. Another technique that does not depend on mean shift is suggested by *Gao & Li (2013)*, which uses an iterative scheme to label events for the segmentation. Furthermore, segmentation methods are improved by applying active contour tracking algorithms and level set methods (*Gao & Chae, 2010*). It shows higher accuracy and visualization of tooth regions as compared to other methods.

## Conventional machine learning algorithms for dental image analysis

Development in the field of Machine Learning (ML) and Artificial Intelligence (AI) is growing over the last few years. ML and AI methods have made a meaningful contribution to the field of dental imaging, such as computer-aided diagnosis & treatment, X-ray image interpretation, image-guided treatment, infected area detection, and information representation adequately and efficiently. The ML and AI make it easier and help doctors diagnose and presume disease risk accurately and more quickly in time. Conventional machine learning algorithms for image perception rely exclusively on expertly designed features, *i.e.*, identifying dental caries involves extracting texture features—an overview of various machine learning algorithms is given in Fig. 7.

ML datasets are generally composed of exclusive training, validation, and test sets. It determines system characteristics by evaluating and testing the dataset then validates the features acquired from the input data. Using the test dataset, one might finally verify ML's precision and extract valuable features to formulate a powerful training model. Table 4 reveals the conventional machine-learning algorithms used for dental X-ray imaging.

## Deep learning techniques for dental image analysis

Artificial intelligence, machine learning, and deep learning approaches assist medical imaging technicians in spotting abnormalities and diagnosing disorders in a fraction of the time required earlier (and with more accurate tests generally). Deep learning (DL) is an improvement of artificial neural networks (ANN), which has more layers and allows for more accurate data predictions (*LeCun, Bengio & Hinton, 2015*; *Schmidhuber, 2015*).

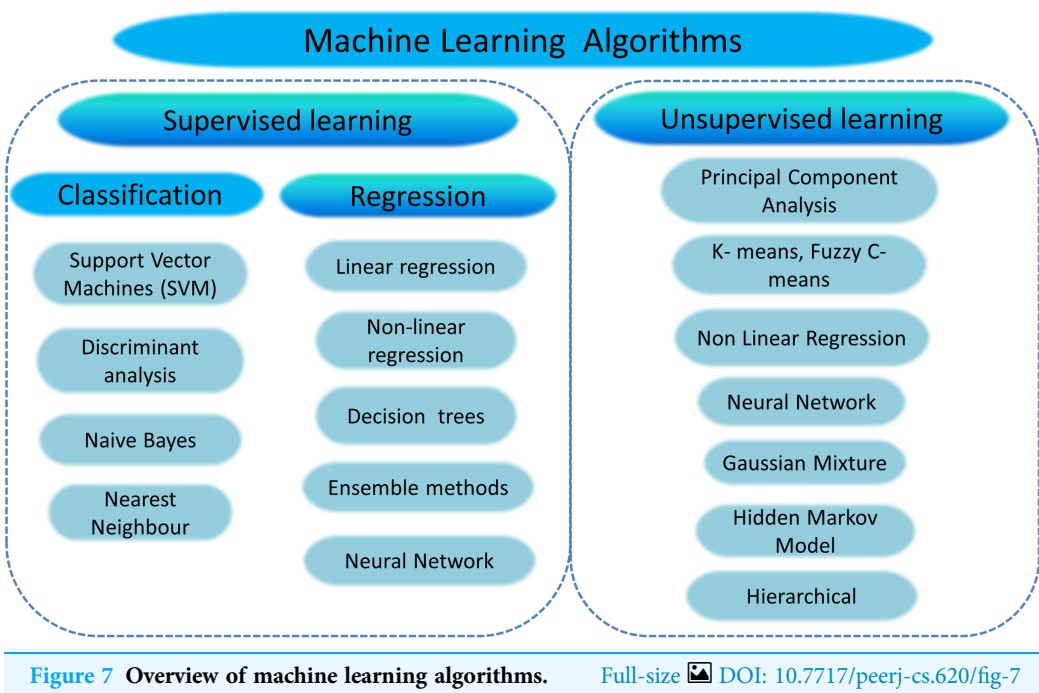

**Figure 7  Overview of machine learning algorithms.**

Deep learning is associated with developing self-learning back-propagation techniques that incrementally optimize data outcomes and increase computing power. Deep learning is a rapidly developing field with numerous applications in the healthcare sector. The number of available, high-quality datasets in ML and DL applications plays a significant role in evaluating the outcome accuracy. Also, information fusion assists in integrating multiple datasets and their use of DL models to enhance accuracy parameters. The predictive performance of deep learning algorithms in the medical imaging field exceeds human skill levels, transforming the role of computer-assisted diagnosis into a more interactive one (*Burt et al., 2018*; *Park & Park, 2018*).

Health diagnostic computer-aided software is used in the medical field as a secondary tool, but developing traditional CAD systems tend to be very strenuous. Recently, there have been introducing deep learning approaches to CAD, with accurate outcomes for different clinical applications (*Cheng et al., 2016*). The research study mostly used a convolution neural network model to analyze other dental imaging modalities. CNN's are a typical form of deep neural network feed-forward architectures, and they are usually used for computer vision and image object identification tasks. CNN's were initially released about two decades back; however, in 2012, AlexNet's architecture outpaced added ImageNet large-scale competition challenges (*Krizhevsky, Sutskever & Hinton, 2012*). Machine vision came in as the deep learning revolution, and since then, CNNs have been rapidly evolving. Feature learning methods have taken a massive turn since the CNN model has come into the picture. Fully convolution neural network Alexnet architecture is used to categorize teeth, including molar, premolar, canine, and incisor, by training cone-beam CT images (*Miki et al., 2017a*; *Oktay, 2017*). *Tuzoff et al. (2019)* applied the Faster R-CNN model, which interprets pipeline and optimizes computation to detect the

**Table 4** The table shows relevant review findings of conventional machine learning algorithms for different imaging modalities.

| Author & Year | Relevant review findings | Images | Feature classifier | Detection |
|---|---|---|---|---|
| **Imaging modality: Periapical X-rays** | | | | |
| (Li et al., 2005, 2007) | To segment the dental Image into normal, abnormal, and potentially abnormal areas, the variational level set function is used. | 60 X-rays | Trained SVM is used to characterize the normal and abnormal regions after Segmentation. | Bone loss & root decay |
| **Imaging modality: Panoramic X-rays** | | | | |
| (Pushparaj et al., 2013) | The geometrical features are used to classify both premolar and molar teeth, while for tooth numbering, the matching templates method is used effectively. | N.A | Feature extraction (Projected principal edge distribution (PPED) + Geometric properties + Region descriptors) + SVM | Teeth numbering and Classification are used to help Forensic odontologists. |
| (Sornam & Prabhakaran, 2017) | The Linearly Adaptive Particle Swarm algorithm is developed and implemented to improve the accuracy rate of the neural system classifier. | N.A | Back Propagation Neural Network (BPNN) and Linearly Adaptive Particle Swarm Optimization (LA-PSO) | Caries detection |
| (Bo et al., 2017) | A two-stage SVM model was proposed for the Classification of osteoporosis. | Dataset consists of 40 images | HOG (histogram of oriented gradients + SVM | Osteoporosis detection |
| (Vila-Blanco, Tomás & Carreira, 2018) | Segmentation of mandibular teeth carried out by applying Random forest regression-voting constrained local model (RFRV-CLM) in two steps: The 1st step gives an estimate of individual teeth and mandible regions used to initialize search for the tooth. In the second step, the investigation is carried out separately for each tooth. | Training images: 261 Testing images: 85 | (RFRV-CLMs) | Adult age teeth detection or a missing tooth for person identification. |
| **Imaging Modality: Photographic color images** | | | | |
| (Fernandez & Chang, 2012) | Teeth segmentation and classification of teeth palate using ANN gives better results as compared to SVM. It shows that ANN is seven-times faster than SVM in terms of time | N.A | ANN + Multilayer perceptrons trained with the error back-propagation algorithm. | Oral infecto-contagious diseases, |
| (Prakash, Gowsika & Sathiyapriya, 2015) | The prognosticating faults method includes the following stages: pre-processing, Segmentation, features extraction, SVM classification, and prediction of diseases. | N.A | Adaptive threshold + Unsupervised SVM classifier | Dental defect prediction |
| **Imaging modality: CBCT or CT** | | | | |
| (Yilmaz, Kayikcioglu & Kayipmaz, 2017) | Classifier efficiency improved by using the forward feature selection algorithm to reduce the size of the feature vector. The SVM classifier performs best in classifying periapical cyst and keratocystic odontogenic tumor (KCOT) lesions. | A total of 50 CBCT 3D scans | Order statistics (median, standard deviation, skewness, kurtosis, entropy) and 3D Haralick Features + SVM | Periapical cyst and keratocystic odontogenic tumor |
| **Imaging modality: Hybrid dataset images** | | | | |
| (Nassar & Ammar, 2007) | A hybrid learning algorithm is used to evaluate the binary bayesian classification filters' metrics and the class-conditional intensities. | Bitewing & Periapical films | Feature extraction + Bayesian classification. | Teeth are matching for forensic odontology. |

(Continued)

| Author & Year | Relevant review findings | Images | Feature classifier | Detection |
|---|---|---|---|---|
| (*Avuçlu & Bacsçiftçi, 2019*) | First, image pre-processing and segmentation are applied to extract the features and quantitative information obtained from the feature extraction from teeth images. Subsequently, features are taken as input to the multilayer perceptron neural network. | A total of 1,315 Dental X-ray images,162 different age groups | Otsu thresholding + Feature extraction (average absolute deviation) + Multilayer perceptron neural network | Age and gender classification |

tooth (*Ren et al., 2017*) and VGG-16 convolutional architecture for classification (*Simonyan & Zisserman, 2014*). These methods are beneficial in practical applications and further investigation of computerized dental X-ray image analysis.

In DXRI, CNNs have been extensively used to detect tooth fractures, bone loss, caries detection, periapical lesions, or also used for the analysis of different dental structures (*Lee et al., 2018b*; *Schwendicke et al., 2019*). Neural networks need to be equipped and refined, and X-ray dataset repositories are necessary (*Lee et al., 2018a*). In *Lee et al. (2019)*, the mask R-CNN model is applied based on a CNN that can identify, classify, and mask artifacts in an image. A mask R-CNN mask operated in two steps. In the first step, the Region of interest (ROIs) selection procedure was performed. Next, the R-CNN mask includes a binary mask similarity to the classification and bounding box foresight for each ROI (*Romera-Paredes & Torr, 2016*; *He et al., 2017*).

Dental structures (enamel, dentin, and pulp) identified using U-net architecture show the best outcome (*Ronneberger, Fischer & Brox, 2015*). CNN is a standard technique for multi-class identification and characterization, but it requires extensive training to achieve a successful result if used explicitly. In the medical sphere, the lack of public data is a general problem because of privacy. To address this issue, (*Zhang et al., 2018*) suggested a technique that uses a label tree to assign multiple labels to each tooth and decompose a task that can manage data shortages. Table 5 presents various studies considering deep learning-based techniques in the field of dentistry.

## Challenges and future directions

After reviewing various works focusing on traditional image processing techniques, it has been perceived that researchers faced multiple challenges in the field of DXRI segmentation and analysis, such as intensity variation in the X-ray images, poor image quality due to noise, irregular shape of an object, limitations of capturing devices, proper selection of methodology and lack of availability of datasets. Also, experience severe challenges in automatically detecting abnormalities, root canal infection, and sudden changes in the oral cavity. Since there are different varieties of dental X-ray images, it is hard to find a particular segmentation approach; it all depends on the precise condition of the X-rays. Some articles have used pre-processed digital X-rays that were manually cropped to include the area of interest. Because of inconsistencies in the manual method, it is hard to accurately interpret and compare outcomes (*Lee, Park & Kim, 2017*).

**Table 5 The table shows relevant review findings of deep learning algorithms for different imaging modalities.**

| Authors | Deep learning architectures | Detection/Application | Metrics |
|---|---|---|---|
| **Imaging modality: Periapical X-rays** | | | |
| (Prajapati, Nagaraj & Mitra, 2017) | CNN and transfer learning | Dental caries, periapical infection, and periodontitis | Accuracy:- 0.8846 |
| (Yang et al., 2018) | Conventional CNN | Automated clinical diagnosis | F1 score 0.749 |
| (Zhang et al., 2018) | CNN (label tree with cascade network structure) | Teeth detection & classification | Precision:- 0.958, Recall:- 0.961 F-score :- 0.959 |
| (Choi, Eun & Kim, 2018) | Conventional CNN | Caries detection | F1max:- 0.74 with FPs:- 0.88 |
| (Lee et al., 2018b) | GoogLeNet Inception v3 CNN network | Caries and Non-caries | Premolar accuracy (premolar):- 0.89, Accuracy (molar):- 0.88, and Accuracy:- 0.82, AUC (premolar):- 0.917, AUC (molar):- 0.890, and an AUC (Both premolar and molar):- 0.845 |
| (Lee et al., 2018a) | CNN (VGG-19) | Periodontally compromised teeth (PCT) | For premolars, the total diagnostic Accuracy(premolars):- 0.810, Accuracy(molars):- 76.7% |
| (Geetha, Aprameya & Hinduja, 2020) | Back-propagation neural network | Caries detection | Accuracy:- 0.971, FPR:- 0.028, ROC :- 0.987, PRC :- 0.987 with learning rate:- 0.4, momentum:- 0.2 |
| **Imaging modality: Panoramic X-rays** | | | |
| (Oktay, 2017) | AlexNet | Teeth detection and classification | Accuracy (tooth detection):- 0.90 Classification accuracy: Molar :-0.9432, Premolar:- 0.9174, Canine & Incissor:- 0.9247 |
| (Chu et al., 2018) | Deep octuplet Siamese network (OSN) | Osteoporosis analysis | Accuracy:- 0.898 |
| (Wirtz, Mirashi & Wesarg, 2018) | Coupled shape model + neural network | Teeth detection | Precision:- 0.790, Recall:- 0.827 Dice coefficient:- 0.744 |
| (Jader et al., 2018) | Mask R-CNN model | Teeth detection | Accuracy:- 0.98, F1-score:- 0.88, precision:- 0.94, Recall:- 0.84, and Specificity:- 0.99 |
| (Lee et al., 2019) | Mask R-CNN model | Teeth segmentation for diagnosis and forensic identification | F1 score:- 0.875, Precision:- 0.858, Recall:- 0.893, Mean'IoU':- 0.877 |
| (Kim et al., 2019) | DeNTNet (deep neural transfer Network) | Bone loss detection | F1 score:- 0.75, Accuracy:- 0.69. |
| (Tuzoff et al., 2019) | R-CNN | Teeth detection and numbering | Tooth detection (Precision:- 0.9945 Sensitivity:- 0.9941) Tooth numbering (Specificity:- 0.9994, Sensitivity = 0.9800) |
| (Fukuda et al., 2019) | DetectNet with DIGITS version 5.0 | Vertical root fracture | Recall:- 0.75, Precision:- 0.93 F-measure:- 0.83 |
| (Murata et al., 2019) | AlexNet | Maxillary sinusitis | Accuracy:- 0.875, Sensitivity:- 0.867, Specificity:- 0.883, and AUC:- 0.875. |
| (Kats et al., 2019) | ResNet-101 | Plaque detection | Sensitivity:- 0.75, Specificity:- 0.80, Accuracy:- 0.83, AUC:- 0.5 |
| (Singh & Sehgal, 2020) | 6-Layer DCNN | Classification of molar, premolar, canine and incisor | Accuracy (augmented database):- 0.95, Accuracy (original database):- 0.92 |

(Continued)

| Authors | Deep learning architectures | Detection/Application | Metrics |
|---|---|---|---|
| (Muramatsu et al., 2020) | CNN (Resnet 50) | Teeth detection and classification | Tooth detection sensitivity:- 0.964 Average classification accuracy (single model):- 0.872, (multisized models):- 0.932 |
| (Banar et al., 2020) | Conventional CNN | Teeth detection | Dice score:- 0.93, accuracy:- 0.54, a MAE:- 0.69, and a linear weighted Cohen's kappa coefficient:- 0.79. |
| **Imaging modality: Bitewing X-rays** | | | |
| (Srivastava et al., 2017) | Fully convolutional neural network FCNN | Detection of dental caries | Recall:- 80.5, Precision:- 61.5, F-score:- 70.0 |
| **Imaging modality: CT & CBCT** | | | |
| (Miki et al., 2017a) | AlexNet architecture | A total of seven-Tooth-type classification (canine, molar, premolar, etc.) | Accuracy:- 0.91 |
| (Miki et al., 2017b) | AlexNet | Teeth detection and classification | Detection accuracy:- 0.774, Classification accuracy:- 0.771 |
| (Hatvani et al., 2018) | Subpixel network + U-Net architecture | Teeth resolution enhancement | Mean of difference (area mm$^2$):- 0.0327 Mean of difference(micrometer):- 114.26 Dice coefficient:- 0.9101 |
| (Torosdagli et al., 2018) | CNN (a long short-term memory (LSTM) network) | Anatomical Landmarking | DSC:- 0.9382 |
| (Egger et al., 2018) | CNN (VGG16, FCN) | Mandible detection | Accuracy:- 0.9877, Dice coefficient:- 0.8964 and Standard deviation:- 0.0169 |
| (Hiraiwa et al., 2019) | AlexNet and GoogleNet | Classification of root morphology (Single or extra) | Diagnostic accuracy:- 0.869 |
| **Imaging modality: Hybrid dataset** | | | |
| (Wang et al., 2016) | U-net architecture (Ronneberger, Fischer & Brox, 2015) | Landmark detection in cephalometric radiographs and Dental structure in bitewing radiographs. | F-score => 0.7 |
| (Lee, Park & Kim, 2017) | LightNet and MatConvNet | Landmark detection | N.A |
| (Karimian et al., 2018) | Conventional CNN | Caries detection | Sensitivity:- 97.93%~99.85% Specificity:- 100% |
| **Imaging modality: Color images/Oral images** | | | |
| (Rana et al., 2017) | Conventional CNN | Detection of inflamed and healthy gingiva | precision:- 0.347, Recall: 0.621, AUC:- 0.746 |
| **Image type not defined** | | | |
| (Imangaliyev et al., 2016) | Conventional CNN | Dental plaque | F1-score:- 0.75 |

Moreover, convolutional neural networks (and their derivatives) are performing outstandingly in dental X-ray image analysis. One notable conclusion is that many researchers use almost the same architectures, the same kind of network, but have very different outcomes. Deep neural networks are most successful when dealing with a large training dataset, but large datasets are not publically available in the DXRI and are not annotated. If vast publicly accessible dental X-ray image datasets were constructed, our research community would undoubtedly benefit exceedingly.

For the future perspective, the dental X-ray image public repository needs to be developed, and data uniformity is required for deep learning applications in dentistry. Also, DXRI aims to create a classifier that can classify multiple anomalies, caries classes, types of jaw lesions, cyst, root canal infection, etc., in dental images using features derived from the segmentation results. There is also a need to build machine learning-based investigative methods and rigorously validate them with a large number of dental professionals. The participation of specialists in this process will increase the likelihood of growth and development. Currently, there exists no universally acceptable software or tool for dental image analysis. However, such a tool is essentially needed to improve the performance of CAD systems and better treatment planning.

## PERFORMANCE MEASURES

In general, if the algorithm's efficiency is more significant than other algorithms, one algorithm is prioritized over another. Evaluating the effectiveness of a methodology requires the use of a universally accessible and valid measure. Various performance metrics have been used to compare algorithms or machine learning approaches depending on the domain or study area. It comprises accuracy, Jaccard index, sensitivity, precision, recall, DSC, F-measure, AUC, MSE, error rate, etc. Here, we include a thorough analysis of the success metrics employed in dental image analysis.

### Performance metrics used for dental image processing

Calculating performance metrics used for dental segmentation is performed by authenticating pixel by pixel and analyzing the segmentation results with the gold standard. Manual annotation of X-ray images done by a radiologist is considered to be the gold standard. Pixel-based metrics are measured using precision, dice coefficient, accuracy, specificity, and F-score widely used in segmentation analysis. Some of the problems in analyzing image segmentation are metric selection, the use of multiple meanings for some metrics in the literature, and inefficient metric measurement implementations that lead to significant large volume dataset difficulties. Poorly described metrics can result in imprecision conclusions on state-of-the-art algorithms, which affects the system's overall growth. Table 6 presents an overview of performance metrics widely used by researchers for dental image segmentation and analysis.

The significance of accuracy and assurance is essential in the medical imaging field. Also, the validation of segmentation achieves the result and dramatically increases the precision, accuracy, conviction, and computational speed of segmentation. Segmentation methods are especially helpful in computer-aided medical diagnostic applications where the interpretation of objects that are hard to differentiate by human vision is a significant component.

### Confusion matrix

The confusion matrix is used to estimate the performance of medical image segmentation and classification. The confusion matrix helps identify the relationship between the outcomes of the predictive algorithm and the actual ones. Some of the terms used for the

**Table 6 Performance metrics used by various researchers for the dental image analysis.**

| Metrics | Symbol | Author's |
|---|---|---|
| True positive rate (sensitivity, recall) | TPR | (Hosntalab et al., 2010; Mortaheb, Rezaeian & Soltanian-Zadeh, 2013; Pushparaj et al., 2013; Ghaedi et al., 2014; Abdi, Kasaei & Mehdizadeh, 2015; Berdouses et al., 2015; Datta & Chaki, 2015b; Alsmadi, 2018; Datta, Chaki & Modak, 2019, 2020) |
| True negative rate (specificity) | TNR | (Hosntalab et al., 2010; Mortaheb, Rezaeian & Soltanian-Zadeh, 2013; Ghaedi et al., 2014; Abdi, Kasaei & Mehdizadeh, 2015; Berdouses et al., 2015; Datta & Chaki, 2015b; Alsmadi, 2018; Datta, Chaki & Modak, 2019) |
| Positive predictive value (precision) | PPV | (Hosntalab et al., 2010; Mortaheb, Rezaeian & Soltanian-Zadeh, 2013; Pushparaj et al., 2013; Berdouses et al., 2015; Datta, Chaki & Modak, 2020) |
| Jaccard index | JAC | (Ji, Ong & Foong, 2014) |
| Dice coefficient | DSC | (Ji, Ong & Foong, 2014; Abdi, Kasaei & Mehdizadeh, 2015; Datta, Chaki & Modak, 2019; Devi, Banumathi & Ulaganathan, 2019) |
| F-Measure (F1 Measure = Dice) | FMS | (Berdouses et al., 2015; Datta, Chaki & Modak, 2020) |
| Accuracy | ACC | (Huang & Hsu, 2008; Olsen et al., 2009; Banu et al., 2014; Nuansanong, Kiattisin & Leelasantitham, 2014; Ghaedi et al., 2014; Lin et al., 2014; Datta & Chaki, 2015a,b; Poonsri et al., 2016; Rad et al., 2018; Osterloh & Viriri, 2019; Datta, Chaki & Modak, 2019, 2020; Devi, Banumathi & Ulaganathan, 2019; Kumar, Bhadauria & Singh, 2020) |
| Mahalanobis distance | MHD | (Pushparaj, Gurunathan & Arumugam, 2013) |
| Hausdorff distance | HD | (Abdi, Kasaei & Mehdizadeh, 2015) |
| Distance vector | DV | (Prajapati, Desai & Modi, 2012) |
| Similarity measure | SM | (Pushparaj, Gurunathan & Arumugam, 2013; Alsmadi, 2018; Singh & Agarwal, 2018) |
| The area under ROC curve | AUC | (Nuansanong, Kiattisin & Leelasantitham, 2014) |
| Cohens kappa coefficient | KAP | (Berdouses et al., 2015) |
| Mean absolute error | MAE | (Vijayakumari et al., 2012; Amer & Aqel, 2015; Tuan et al., 2018; Kumar, Bhadauria & Singh, 2020) |
| Mean square error | MSE | (Vijayakumari et al., 2012; Singh & Agarwal, 2018; Tuan et al., 2018) |
| Error rate | ERR | (Zhou & Abdel-Mottaleb, 2005; Nomir & Abdel-Mottaleb, 2008; Hosntalab et al., 2010; Harandi & Pourghassem, 2011; Lira et al., 2014; Datta & Chaki, 2015b; Purnama et al., 2015; Tuan et al., 2018; Banday & Mir, 2019) |
| Failure rate | FR | (Said et al., 2006; Al-sherif, Guo & Ammar, 2012) |

confusion matrix are given in Table 7: True positive (TP): correctly identified or detected; False positive (FP): evaluated or observed incorrectly; False negative (FN): wrongly rejected; True Negative (TN): correctly rejected. In the approach (*Mahoor & Abdel-Mottaleb, 2005*), experimental outcomes proved that molar classification is relatively easy compared to premolars, and for teeth classification, centroid distance is less effective than a coordinate signature. Various metrics such as the signature vector, force field (FF), and Fourier descriptor (FD) were used to test the efficiency of the approach given by *Nomir & Abdel-Mottaleb (2007)*, and for matching euclidean distance and absolute distance, FF & FD give small values, suggesting that they performed better than the others. Here, FF & FD give small values for matching Euclidean distance and absolute distance, indicating that the performance is better than the other two methods. In another approach (*Prajapati, Desai & Modi, 2012*), feature vectors are evaluated and used to find the image distance vector ($D_n$) using formula: $D_n = \sum |T_n FV - FVQ|$, where feature vector (TnFV) is used for database image and (FVQ) is used for the query image.

**Table 7 Confusion matrix.**

| | |
|---|---|
| *True positive rate (Recall/Sensitivity)*: It implies how the caries lesion is accurately detected when it is present there. | *Sensitivity* is expressed as $\dfrac{TP}{TP + FN}$ |
| *True negative rate (Specificity)*: That is the amount of negative caries lesion examination when there's no affected lesion. | *Specificity* is measured as $\dfrac{TN}{TN + FP}$ |
| *Dice Coefficient*: This metric measures between two samples. | It is defined as $\dfrac{(2\|A \cap B\|)}{(\|A\| + \|B\|)}$, where $\|A\|$ and $\|B\|$ are the number of elements in the sample. |
| *Accuracy*: It can be defined as the percentage of correctly classified instances. | It is calculated as $\dfrac{TP + TN}{TP + TN + FN + FP} * 100$. |
| *Precision:* It explains the pureness of our positive detections efficiently compared to the ground truth. | It is the positive predictive value defined as $\dfrac{TP}{TP + FP}$ |
| *F-Score:* The F-score is a process of combining the model's precision and recall and the harmonic mean of the model's precision and recall. | It is expressed as $2 \times \dfrac{Precision \times Recall}{Precision + Recall}$ |

The minimum value of the distance vector indicates the best match of the image with the database image.

The study (*Huang et al., 2012*) shows better isolation precision accuracy for the segmentation of jaws as compared with Nomir and Abdel–Mottaleb. Another method evaluated the complete length of the tooth and capered with the dentist's manual estimation (*Harandi & Pourghassem, 2011*). Here, measurement error (ME) is evaluated for root canals applying the formula: $ME = \frac{Mesured\ length}{Actual\ length}$ and evaluated ME is lowest for one canal compared to two and three canals.

*Niroshika, Meegama & Fernando (2013)* traced the tooth boundaries using active contour and distance parameters are compared with the Kass algorithm. The value of the standard distance parameter was found to be lower than that of the Kass algorithm, implying that the proposed method is more efficient for tracing the tooth boundary than the Kass algorithm. Another approach used for counting molar and premolar teeth is considering precision and sensitivity (*Pushparaj et al., 2013*). Here performance is using metric $'\eta'$ is given by: $\eta = \frac{(m-n)}{n} * 100$. Where 'm' represents the total number of teeth counted, and 'n' represents the incorrectly numbered teeth. The counting of molar and premolar teeth is more than 90% accurate using this method.

In *Abdi, Kasaei & Mehdizadeh (2015)*, mandible segmentation and Hausdorff distance parameters were compared to the manually annotated gold standard. The algorithm results appear to be very close to the manually segmented gold standard in terms of sensitivity, accuracy, and dice similarity coefficient (DSC). In *Amer & Aqel (2015)*, a wisdom tooth is extracted, and the mean absolute error (MAE) is used to equate the procedure with the other two methods. As compared to other approaches, the lower MAE value showed better segmentation.

In *Poonsri et al. (2016)*, precision is calculated for single-rooted and double-rooted teeth using template matching. According to their study, segmentation accuracy is greater than 40%. *Son & Tuan (2016, 2017)* used the following cluster validity metrics: PBM, Simplified Silhouette Width Criterion (SSWC), Davis-Bouldin (DB), BH, VCR, BR, and

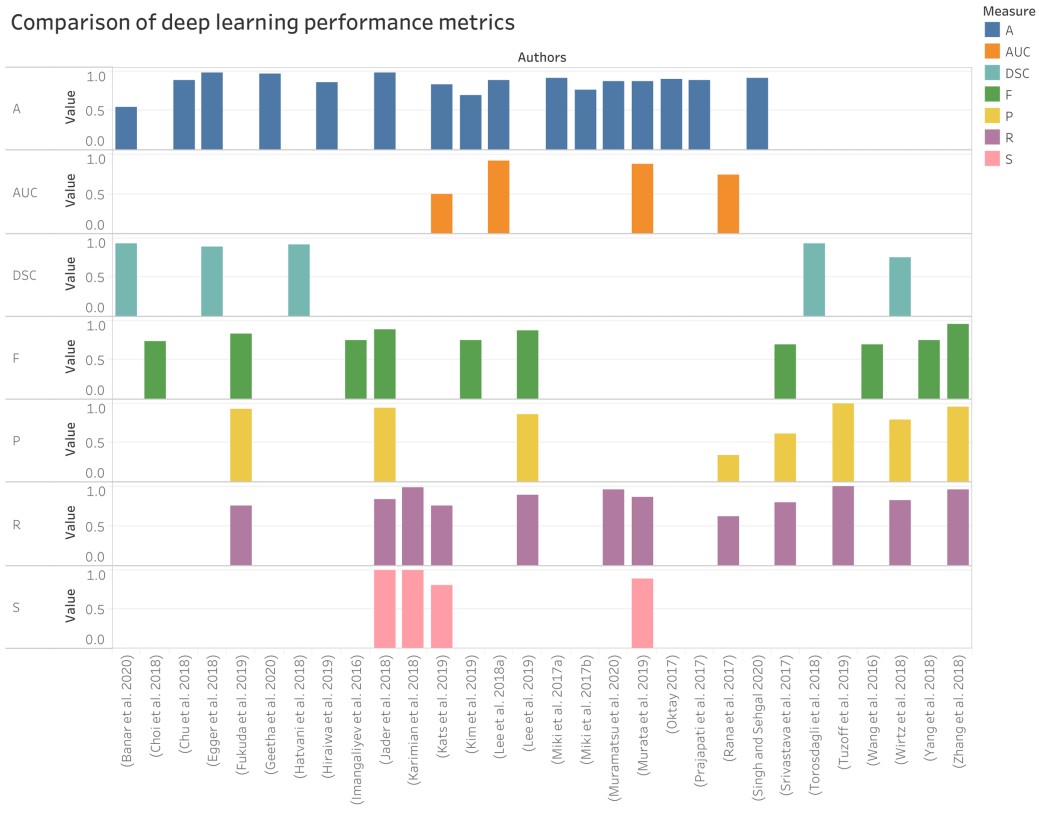

**Figure 8** Performances measure comparisons used for deep learning methods.

TRA, and the measures of these parameters indicate the best performance as compared with the results of current algorithms.

*PBM:* The maximum value of this index is said to be the PBM index, across the hierarchy provides the best partitioning.

*Simplified Silhouette Width Criterion (SSWC):* The silhouette analysis tests how well the observation is clustered and calculates the average distance between clusters. The silhouette plot shows how similar each point in a cluster is to the neighboring clusters' points.

*Davies-Bouldin index (DB):* This index determines the average 'similarity' amongst clusters, in which the resemblance is a metric that measures the distance between clusters with the size of clusters themselves. The lower Davies-Bouldin index refers to a model with a greater detachment of clusters.

*Ball and Hall index (BH):* It is used to determine the distance within a group, with a higher value showing better results.

*Calinski-Harabasz index*, also called Variance Ratio Criterion (VCR): It can be applied to evaluate the partition data by variance, and its higher value indicates good results.

*Banfeld-Raftery index (BR):* It is evaluated using a variance-covariance matrix for each cluster.

**Table 8 Dental X-ray image dataset description used for deep learning methods.**

| Authors & Year | Dataset description |
| --- | --- |
| (Eun & Kim, 2016) | Periapical X-rays: 500 periapical images used for training where each Image is containing five teeth and 100 images used for testing with corresponding ground truth. |
| (Wang et al., 2016) | Total number of patients: 400 (100 additional patients)<br>Cephalometric radiographs: 400 images .tiff format dimension of 1,935 × 2,400 pixels, 120 bitewing radiographs (new) (Age group 6 to 60 yrs)<br>Software used: Soredex CRANEXr Excel Ceph machine (Tuusula, Finland) and Soredex SorCom software (3.1.5, version 2.0) |
| (Prajapati, Nagaraj & Mitra, 2017) | Periapical RadioVisio Graphy (RVG) X-ray: 251 images (labeled dataset) where 180 used for training, 26 images for testing & 45 images validation. |
| (Rana et al., 2017) | Color images: Training and validation data consist of 258 images & 147 images. |
| (Lee, Park & Kim, 2017) | A total of 300 Dental X-ray images with resolution 1,935 × 2,400 pixels and 150 images used for training, and 150 images used for testing. |
| (Srivastava et al., 2017) | Bitewing images: More than 3,000 images |
| (Miki et al., 2017a) | CBCT dataset taken from Asahi University Hospital, Gifu, Japan.<br>Two dental units: Veraviewepocs 3D (J.Morita Mfg, Corp., Kyoto, Japan) and Alphard VEGA (Asahi Roentgen Ind. Co., Ltd., Kyoto, Japan |
| (Miki et al., 2017b) | CT data: 52 images, Training group: 42 images, testing group: 10 images |
| (Oktay, 2017) | Panoramic Images: Dataset taken from 3-different X-ray machines have image dimensions 2871 × 1577, 1435 × 791, or 2612 × 1244 pixels. Testing and validation are done using images of 100 different people. |
| (Yang et al., 2018) | A small dataset of 196 periapical images used, and also augmentation is performed. |
| (Zhang et al., 2018) | Periapical Images: Initially for training, 800 images and 200 used for testing.<br>and data is annotated with the help of bounding box labels in 32 teeth position. |
| (Wirtz, Mirashi & Wesarg, 2018) | Panoramic X-rays: 14 test images used.<br>Image augmentation is used to increase training images up to 4000. |
| (Choi, Eun & Kim, 2018) | Periapical X-rays: 475 images dimension of 300 × 413 from 688 × 944 or 1200 × 1650. |
| (Jader et al., 2018) | Panoramic X-ray images:<br>A total of 193 images used for training containing 6987 teeth and 83 images for validation containing 3,040. |
| (Lee et al., 2018b) | Periapical Images: 3,000 images .jpeg format dimension resized to 299 × 299 pixels The training and validation dataset was 2,400 and a test dataset of 600. The training and validation dataset consisted of 1,200 dental caries and 1,200 non-dental caries, and the test dataset consisted of 300 dental caries and 300 non-dental caries. Augmentation is done up to ten times for training. |
| (Hatvani et al., 2018) | Micro CT images: a training set consists of 5,680 slices and a test set of 1,824 slices was used. |
| (Torosdagli et al., 2018) | CBCT dataset of 50 patients and qualitative visual inspection were done for 250 patients with high variability. |
| (Karimian et al., 2018) | Training is performed using different batches containing ten optical coherence tomography (OCT) images per batch. |
| (Lee et al., 2018a) | Periapical X-ray images resized to 224 × 224 pixels (from the original 1,440 × 1,920 pixels) in .png format : For training ($n = 1,044$), validation ($n = 348$), and test ($n = 348$) datasets. |
| (Egger et al., 2018) | CT dataset containing 45 images as DICOM files with dimension 512 × 512 from a department of craniomaxillofacial surgery in Austria. 1st Image set containing 1,680 slices, 2nd one with induced noise images 6720 slices, 3rd after transformation 13,440 slices, and 4th covered augmentation 18,480 slices |
| (Chu et al., 2018) | Panoramic X-ray: 108 images. |
| (Hiraiwa et al., 2019) | CBCT images and panoramic radiographs used for 760 mandibular first molars (400 patients) |
| (Lee et al., 2019) | Panoramic X-rays: Dimensions of 2,988 × 1,369 pixels.<br>Total 846 annotated tooth images.<br>Training group: 30 radiographs, Validation & testing: 20 images.<br>Augmentation technique used to reduce overfitting ( obtained 1,024 training samples from 846 original data points ) |
| (Kim et al., 2019) | Panoramic Images:12,179 images (annotated by experts)<br>Trained, validated, and tested using 11, 189, 190, and 800. |

(Continued)

| Table 8 (continued) | |
|---|---|
| **Authors & Year** | **Dataset description** |
| (*Tuzoff et al., 2019*) | Panoramic radiographs: 1,352 images<br>Training group: 1,352 images, Testing group: 222 images |
| (*Murata et al., 2019*) | Panoramic X-rays: Total patients: 100 (50 men and 50 women), Training data for 400 healthy and 400 inflamed maxillary sinuses and data augmentation used to make 6,000 samples |
| (*Kats et al., 2019*) | Panoramic X-ray:65 images and augmentation performed. |
| (*Fukuda et al., 2019*) | Panoramic X-ray: 300 images with 900 × 900 pixels.<br>Training set: 240 images, Testing set: 60 images |
| (*Singh & Sehgal, 2020*) | Panoramic X-rays: Total 400 images. Training group: 240 images, Testing group: 160 images. Also, augmentation is done by using transformation. |
| (*Muramatsu et al., 2020*) | Panoramic X-rays: 100 images dimension of 3,000 × 1,500 pixels used for testing and training both. |
| (*Geetha, Aprameya & Hinduja, 2020*) | Periapical X-rays: 105 images saved as in .bmp format dimension resized to 256 × 256, where caries identified 49 images<br>Training, validation, and testing consists of 49 caries and 56 sound dental X-ray images. |
| (*Banar et al., 2020*) | Panoramic (OPGs) image dataset of 400 images used. |

*Difference-like index (TRA):* It calculates the cluster difference, and a higher value gives the best results.

Comparison of various performance metrics used in dental X-ray imaging considering deep learning methods are given in Fig. 8.

## DATASET DESCRIPTION

The researcher in the dental imaging field has used various types of databases. In which some of the databases are available online, while some records are not present. The most prominent dilemma is finding out which investigation has given valid results because everyone has shown promising results on their datasets. All the dental imaging databases that have been used so far are given in Table 8.

## CONCLUSION

Dental X-ray image analysis is a challenging area, and it receives significantly less attention from the community of researchers. There is, however, no systematic review that addresses the state-of-the-art approaches of DXRI. This paper has performed a thorough analysis of more than 130 techniques suggested by different researchers over the last few decades. This study presented a survey of various segmentation and classification techniques widely used for dental X-ray imaging. Methods are characterized as image processing, conventional machine learning, and deep learning. Furthermore, a novel taxonomy mainly focusing on the imaging modalities-based categorization such as bitewing, periapical, panoramic, CBCT/CT, hybrid datasets, and color pictures. Various studies have found that opting for one type of segmentation technique is very difficult in conventional image-processing methods because of image dataset variability. The primary barrier in the growth of a high-performance classification model is the requirement of an annotated datasets, as pointed by various researchers mentioned in this study. Dental X-ray imaging data is not the same as other medical images because of the different image

characteristics. This difference has an impact on the deep learning model's adaptability during image classification. It is also challenging to validate and verify the algorithm's correctness because of the inadequate datasets available for the hypothesis.

Now we would like to bring the researcher's attention towards future directions in DXRI. Since most dental X-ray image analysis methods result in decreased efficiency, more sophisticated segmentation techniques should be designed to improve clinical treatment. Further, it is being observed that limited work is employed in the recent studies to detect caries classes such as classes I–VI, and root canal infection. Researchers should therefore focus on implementing new methodologies for caries classification and detection. Recently, deep learning has improved DXRI segmentation and classification performance and requires large annotated image datasets for training, but large annotated X-ray datasets are not publicly accessible. Further, a public repository for dental X-ray images needs to be developed. It is still an open problem so that we can expect new findings and research outcomes in the coming years.

### Funding
The authors received no funding for this work.

### Competing Interests
The authors declare that they have no competing interests.

### Author Contributions
- Anuj Kumar conceived and designed the experiments, performed the experiments, analyzed the data, performed the computation work, prepared figures and/or tables, authored or reviewed drafts of the paper, and approved the final draft.
- Harvendra Singh Bhadauria conceived and designed the experiments, performed the experiments, analyzed the data, performed the computation work, prepared figures and/or tables, authored or reviewed drafts of the paper, and approved the final draft.
- Annapurna Singh conceived and designed the experiments, performed the experiments, analyzed the data, performed the computation work, prepared figures and/or tables, authored or reviewed drafts of the paper, and approved the final draft.

### Data Availability
This is a Review Article; there are no data or code files.

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
