# Peer review of "Descriptive analysis of dental X-ray images using various practical methods: A review"

_PeerJ Computer Science, doi:10.7717/peerj-cs.620_

## Round 0.1 · original submission · Major Revisions

Dear Dr. Kumar,

Thank you for your submission to PeerJ Computer Science.
It is my opinion as the Academic Editor for your article - Descriptive analysis of dental X-ray images using various practical methods: A review - that it requires a number of Major Revisions.

My suggested changes and reviewer comments are shown below and on your article 'Overview' screen.

Please address these changes and resubmit.

Reviewer 1 ·

Basic reporting

The authors presented a comprehensive survey on dental x-ray imaging.

Experimental design

Can be improved.

Validity of the findings

Not applicable.

Additional comments

1. Is it the first of it kind survey on dental x-ray imaging? If not compare the current survey with existing surveys and highlight how it is different or enhances the existing surveys.
2. Some of the recent works on application of DN/CNN on several domains such as the following can be discussed when discussing about ML algorithms. "Deep learning and medical image processing for coronavirus (COVID-19) pandemic: A survey, A novel PCA–whale optimization-based deep neural network model for classification of tomato plant diseases using GPU, Hand gesture classification using a novel CNN-crow search algorithm".
3. A good survey should present a detailed section on challenges faced by the existing methodologies and also give directions to the researchers who want to carry out the research in that domain. I suggest that the authors can add a section, "challenges and future directions".
4. A good survey paper should have eye catching images. Check this paper for reference "Deep learning and medical image processing for coronavirus (COVID-19) pandemic: A survey"

·

Basic reporting

Paper is well organized.

Experimental design

Rigorous investigation performed to a high technical & ethical standard

Validity of the findings

Conclusions are well stated, linked to original research question & limited to supporting results.

Additional comments

Please check grammar to submit final paper.

Reviewer 3 ·

Basic reporting

no comment

Experimental design

A review method was chosen as research design

Validity of the findings

Appropriate

Additional comments

The authors have chosen a very important topic. The paper Is very interesting and timely. I suggest the authors incorporate these changes to further improve the quality of the paper.

1. There are many Typos and grammatical errors throughout the manuscript.
2. Figures clarity should be thoroughly enhanced.
3. The introduction lacks many important references in the field.
4. I wish to reorganize your paper with better and more clarity by Rectifying the above comments and submit the revised version.
5. Introduction and related work is not synchronized.
6. The conclusion and future work part can be extended to have a better understanding of the approach and issues related to that which can be taken into consideration for future work.

Please cite the following relevant literature but not limited to:
1) Rehman, Z. U., Zia, M. S., Bojja, G. R., Yaqub, M., Jinchao, F., & Arshid, K. (2020). Texture based localization of a brain tumor from MR-images by using a machine learning approach. Medical hypotheses, 141, 109705.
2) Bhattacharya, S., Maddikunta, P. K. R., Pham, Q. V., Gadekallu, T. R., Chowdhary, C. L., Alazab, M., & Piran, M. J. (2021). Deep learning and medical image processing for coronavirus (COVID-19) pandemic: A survey. Sustainable cities and society, 65, 102589.
3) Sai Ambati, L., Narukonda, K., Bojja, G. R., & Bishop, D. (2020). Factors Influencing the Adoption of Artificial Intelligence in Organizations-From an Employee's Perspective.
4) Shabaan, M., Arshad, K., Yaqub, M., Jinchao, F., Zia, M. S., Boja, G. R., ... & Munir, R. (2020). Survey: smartphone-based assessment of cardiovascular diseases using ECG and PPG analysis. BMC medical informatics and decision making, 20(1), 1-16.

---

## Round 0.2 · accepted · Accept

Dear Authors,

I am happy to accept the article based on the feedback received from the reviewers.

Reviewer 1 ·

Basic reporting

In this article, the authors have provided a comprehensive survey of dental image segmentation and analysis by investigating more than 130 research works conducted through various dental imaging modalities, such as various modes of X-ray, CT(Computed Tomography), CBCT (Cone Beam Computed Tomography), etc.

Experimental design

Satisfactory

Validity of the findings

Satisfactory

Additional comments

The authors have addressed all the comments. The manuscript can be accepted for publication.

·

Basic reporting

The authors have addressed my suggestions. I would like to accept this paper.

Experimental design

The authors have addressed my suggestions. I would like to accept this paper.

Validity of the findings

The authors have addressed my suggestions. I would like to accept this paper.

Reviewer 3 ·

Basic reporting

Everything looks good. Authors did a good job to modify the manuscript based on reviewer comments.

Experimental design

NA

Validity of the findings

NA

Additional comments

Good job.